# Moment Matching Q-Learning

**Yiyan (Edgar) Liang** [1]   **Sifei Liu**   **Weitong Zhang** [1]

## Abstract

Score-based and flow-based generative models exhibit remarkable expressive capacity in capturing complex distributions, and have been extensively deployed in tasks ranging from image generation to reinforcement learning. Nevertheless, these models suffer from prolonged inference latency, which imposes a significant computational bottleneck in RL with iterative sampling. To overcome this limitation, we propose a new framework named *Moment Matching Q-Learning* (MoMa QL), which utilizes a technique from statistical hypothesis testing known as maximum mean discrepancy (MMD) that intend to match all orders of statistics between the original and target distribution. By enforcing strong regularization on all moment statistics, this algorithm guarantees distribution-level convergence for conditional score function and remains stable under various hyperparameters. Empirically, we show that our method MoMa QL is more computationally efficient with a comparable if not competitive performance in various D4RL tasks. Remarkably, by accelerating the action sampling process for flow-based policies, MoMa QL demonstrates superior performance in offline-to-online RL tasks because of faster and stronger adaptability for online interactive finetuning.

## 1. Introduction

Offline reinforcement learning (RL) aims to derive an optimal decision-making policy from a previously collected dataset without further environment interaction (Lange et al., 2012). Circumventing risky, costly, and inefficient online interactions, offline RL enables models to fully exploit prior data, and thus gained significant traction in safety-critical

---

[1]School of Data and Information Science, University of North Carolina at Chapel Hill, Chapel Hill, USA. Correspondence to: Weitong Zhang <weitongz@unc.edu>.

*Proceedings of the $43^{rd}$ International Conference on Machine Learning*, Seoul, South Korea. PMLR 306, 2026. Copyright 2026 by the author(s).

applications such as autonomous driving and robotic manipulation (Levine et al., 2020). However, learning exclusively from previously collected data can be more than challenging. Conventional reinforcement learning algorithms typically suffer from distributional shift as they evaluate actions outside the support of the behavior policy, where value estimations are unreliable (Wang et al., 2023). Moreover, as the datasets are growing larger and more diverse, the behavioral distribution is also more complex and multi-modal, which necessitates a more expressive policy class to represent the complex policies (Mandlekar et al., 2022).

To model these complex and potentially multimodal policy distributions, Gaussian mixture model (GMM) (Jacobs et al., 1991; Ren et al., 2021) and variational auto-encoders (VAE) (Kumar et al., 2019) are frequently employed for policy representation, as their tractable sampling and efficient optimization properties allow them to capture the underlying probability distributions of expert behaviors. More advanced frameworks, such as Langevin dynamics-based approaches (Chi et al., 2025) and probability flow methods (Zheng et al., 2023) offer superior expressivity, and thus they have been extensively integrated into both imitation learning and offline RL scenarios. However, the slow sampling speed for these generative models remains unacceptable for various computationally intensive online tasks. Consequently, there is an imperative need for more computationally efficient sampling methods.

To address this problem, we introduced *Moment Matching Q-Learning* (MoMa QL), which involves a stable and theoretically rigorous policy learning procedure based on actor-critic style algorithm. MoMa QL enables actor's policy to operate on the time-dependent marginal distributions of stochastic interpolates (Albergo et al., 2025), which connects two arbitrary probability density functions. By learning a function mapping from any marginal distribution at time $t$ to any marginal at time $s < t$, this framework facilitates a seamless transition across the probability flow trajectory, MoMa QL naturally supports both single- or multi-step action sampling. With this accelerated action sampling, MoMa QL enables a simultaneous critic update through it's policy training procedure. In summary, our contribution can be summarized as follows:

• We introduce MoMa QL, which motivates from minimiz-

ing the Maximum Mean Discrepancy (MMD) of conditional distributions derived from two intermediate diffusion steps $r < s$ to enable the diffusion process to "jump" from $r$ to $s$ via one-step induction.

- Theoretically, we prove that the effectiveness and convergence of the MoMa QL with the fact that the consistency models can be viewed as a special case of our results.

- Empirically, we demonstrated that MoMa QL exhibit superior performance comparing with a series of offline RL methods. Specially, thanks to the accelerated action sampling process, we show that MoMa QL can be efficiently finetuned with online rollouts and outperforms existing offline-to-online RL methods.

## 2. Preliminaries

### 2.1. Offline Reinforcement Learning

We consider the offline reinforcement learning setting in this paper. The environment in RL can be formulated as a Markov decision process $\mathcal{M} := (\mathcal{S}, \mathcal{A}, \mathcal{P}, R, \rho, \gamma)$, with state space $\mathcal{S}$, action space $\mathcal{A}$, environment dynamics $\mathcal{P}(\mathbf{s}'|\mathbf{s}, \mathbf{a}) : \mathcal{S} \times \mathcal{S} \times \mathcal{A} \to I$, reward function $R(\mathbf{s}, \mathbf{a}) : \mathcal{S} \times \mathcal{A} \to \mathbb{R}$, initial distribution $\rho \in \Delta(S)$ and discounted factor $\gamma \in [0, 1)$ (Sutton et al., 1998), where we denote the set $[0, 1]$ as $I$ and the set of probability distributions over a space $\mathcal{X}$ as $\Delta(\mathcal{X})$.

The goal of Offline RL is to learn a policy $\pi_\theta(\mathbf{a}|\mathbf{s}) : \mathcal{S} \to \Delta(\mathcal{A})$, a conditional distribution parametrized by $\theta$, which maximizes the cumulative discounted reward $\mathbb{E}\left[\sum_{t=0}^{H} \gamma^t r(\mathbf{s}_t, \mathbf{a}_t)\right]$ over horizon $H$. This is equivalent to a constrained optimization problem with respect to $\theta$.

In the offline setting, the dataset $\mathcal{D} := \{\tau^n\}_{n=1}^{N}$ is static and doesn't involve environment interaction, where each $\tau^i := (\mathbf{s}_0, \mathbf{a}_0, \ldots, \mathbf{s}_H, \mathbf{a}_H)$ denotes one single trajectory.

In addition to the purely offline setting, we also consider the offline-to-online RL regime, in which we aim to finetune a pre-trained policy via online interactions with the environment after pretraining using offline dataset.

### 2.2. Continuous-time Probabilistic Generative Models

For a given data distribution $q(\mathbf{x}) \in \Delta(\mathbb{R}^d)$, diffusion models (Ho et al., 2020; Song et al., 2020) and flow matching (Liu et al., 2023; Lipman et al., 2023) construct time-dependent variable $x_t$ which can be viewed as an interpolation between sampled data $\mathbf{x} \sim q(\mathbf{x})$ and noise $\boldsymbol{\varepsilon} \sim \mathcal{N}(0, I)$ in the form as $\mathbf{x}_t = \alpha_t \mathbf{x} + \sigma_t \boldsymbol{\varepsilon}$ with $\alpha_0 = \sigma_1 = 1, \alpha_1 = \sigma_0 = 0$. Essentially, the problem can be formulated as solving a *Schrödinger bridge* (Chen et al., 2016; De Bortoli et al., 2021) between $\boldsymbol{\varepsilon}$ and $q(x)$. Within this perspective, Variance Preserved diffusion conventionally chooses $\alpha_t = \cos\left(\frac{\pi}{2}t\right), \sigma_t = \sin\left(\frac{\pi}{2}t\right)$ and flow matching chooses

$\alpha_t = 1 - t, \sigma_t = t$. Both diffusion models and flow matching aim to learn a time-dependent vector field $\mathbf{v}_t(\mathbf{x}) := \frac{\mathrm{d}\mathbf{x}_t}{\mathrm{d}t}$ parameterized by a neural network. For score based diffusion model with deterministic sampling, the vector field can be represented as:

$$\mathbf{v}_t(\mathbf{x}) = \alpha_t - \sigma_t^2 \mathbf{s}_\theta(\mathbf{x}, t)/2,$$

where $\mathbf{s}_\theta(\mathbf{x}_t, t) := \nabla_{\mathbf{x}} \log p_t(\mathbf{x})$ denotes the score function. The model is trained with the objective

$$\mathcal{L}_{\mathrm{DSM}}(\theta) = \mathbb{E}_{t, \mathbf{x}_0, \boldsymbol{\varepsilon}}\left[\|\mathbf{s}_\theta(\mathbf{x}_t, t) + \boldsymbol{\varepsilon}/\sigma_t\|^2\right],$$

where $\boldsymbol{\varepsilon} \sim \mathcal{N}(0, I)$. In contrast, flow matching directly parameterizes the velocity field $\mathbf{v}_t(\mathbf{x})$ and trains the model by regressing toward a target conditional velocity. Given a stochastic interpolation path $\mathbf{x}_t$ between samples from the base and target distributions, the optimal velocity field is defined as $\mathbf{v}_t^\star(\mathbf{x}_t) = \mathbb{E}\left[\frac{\mathrm{d}\mathbf{x}_t}{\mathrm{d}t}|\mathbf{x}_t\right]$, and the corresponding flow matching loss function is

$$\mathcal{L}_{\mathrm{FM}}(\theta) = \mathbb{E}_{t, \mathbf{x}_0, \mathbf{x}_1}\left[\|G_\theta(\mathbf{x}_t, t) - \mathbf{v}_t^\star(\mathbf{x}_t)\|^2\right],$$

where $G_\theta(\mathbf{x}_t, t)$ is a parameterized neural network.

*Stochastic Interpolants* (Albergo et al., 2025) provides a unified and flexible perspective that encapsulates both SDE-based models like diffusion and ODE-based models like flow matching. It constructs a conditional interpolation $q_t(\mathbf{x}_t|\mathbf{x}, \boldsymbol{\varepsilon}) = \mathcal{N}(I_t(\mathbf{x}, \boldsymbol{\varepsilon}), \gamma_t^2 I)$ with constraints $I_0(\mathbf{x}, \boldsymbol{\varepsilon}) = \mathbf{x}$, $I_1(\mathbf{x}, \boldsymbol{\varepsilon}) = \boldsymbol{\varepsilon}$ and $\gamma_0 = \gamma_1 = 0$. The conditional interpolant velocity can thus be represented as $\mathbf{v}_t = \partial_t I_t(\mathbf{x}, \boldsymbol{\epsilon}) + \dot{\gamma}_t \mathbf{z}$ where $\mathbf{z} \sim \mathcal{N}(0, I)$. Similar to flow matching, we can learn a deterministic sampler via $G_\theta(\mathbf{x}_t, t) \approx \mathbb{E}_{\mathbf{x}, \boldsymbol{\varepsilon}, \mathbf{z}}[\mathbf{v}_t|\mathbf{x}_t]$. Given linear interpolation condition $I_t(\mathbf{x}, \boldsymbol{\epsilon}) = \alpha_t \mathbf{x} + \sigma_t \boldsymbol{\epsilon}$, it's easy to show that stochastic interpolants reduces to flow matching when $\gamma_t \equiv 0$ and reduces to diffusion when $\boldsymbol{\varepsilon} \sim \mathcal{N}(0, I)$.

### 2.3. Policy Regularization

Policy regularization adds constraints or penalties to an agent's learning process to prevent it from overfitting and promote stable and robust behavior. Actor-Critic (AC) plays an important role in policy regularization, which casts an explicit constraint on the policy and thus enhances both the stability and the asymptotic performance of the agent. Following conventional behavior-regularized actor-critic framework (BRAC) (Wu et al., 2019; Fujimoto & Gu, 2021), we minimize the empirical risk for both actor and critic by:

$$\mathcal{L}_Q(\phi) = \mathbb{E}_{\substack{\mathbf{s}, \mathbf{a}, r, \mathbf{s}' \sim \mathcal{D}, \\ \mathbf{a}' \sim \pi_\theta}}[(Q_\phi(\mathbf{s}, \mathbf{a}) - r - \gamma Q_\phi^\perp(\mathbf{s}', \mathbf{a}'))^2], \quad (1)$$

$$\mathcal{L}_\pi(\theta) = \mathbb{E}_{\mathbf{s}, \mathbf{a} \sim \mathcal{D}, \mathbf{a}^\pi \sim \pi_\theta}[\underbrace{-\eta Q_\phi(\mathbf{s}, \mathbf{a}^\pi)}_{\text{Q loss}} - \underbrace{\log \pi(\mathbf{a}|\mathbf{s})}_{\text{BC loss}}], \quad (2)$$

where $Q_\phi(\mathbf{s}, \mathbf{a}) : \mathcal{S} \times \mathcal{A} \to \mathbb{R}$ denotes the state-action value function parameterized by $\phi$, $Q_\phi^\perp(\mathbf{s}, \mathbf{a})$ is a target network which stops gradient and delays the update (Mnih et al., 2013) and $\alpha$ controls the strength of the BC regularizer. The critic loss $\mathcal{L}_Q(\phi)$ minimizes the standard Bellman error, while the actor loss $\mathcal{L}_\pi(\theta)$ maximizes values with reparameterized gradients through $\mathbf{a}^\pi$. For the actor, the BC loss is additionally applied to prevent the policy from deviating too much from the behavioral policy's distribution. Despite its simplicity, BRAC is one of the most performant frameworks on standard D4RL tasks, and thus we try to build our algorithm on a variant of the BRAC framework.

## 2.4. Maximum Mean Discrepancy

Consider a reproducing kernel Hilbert space (RKHS) $\mathcal{H}$ of functions $\mathbb{R}^d \to \mathbb{R}$ associated to a bounded positive definite kernel $k$ on $R^d$, the Maximum Mean Discrepancy (MMD) between two probability measures $\mu$ and $\nu$ on $\mathbb{R}^d$ is:

$$\mathsf{MMD}(\mu, \nu) = \sup_{f \in \mathcal{H}, \|f\|_\mathcal{H} \le 1} \left| \int f \, \mathrm{d}\mu - \int f \, \mathrm{d}\nu \right|. \quad (3)$$

MMD is a metric if and only if the kernel mean embedding $\mu \to \int k(x, \cdot)\mu(\mathrm{d}x)$ is *injective*. Kernels which satisfies such a property are called *characteristic*, and one of the most important example is the RBF kernel $k(x, y) = e^{-\frac{||x-y||^2}{2\sigma^2}}$ (Vert et al., 2004). RBF kernel also implies an inner product of infinite-dimensional feature maps consisting of all orders of moments of $\mu$ and $\nu$, which can be respectively denoted and $\mathbb{E}_\mu(\mathbf{x}^j)$ and $\mathbb{E}_\nu(\mathbf{x}^j)$ for any $j \ge 1$ (Suthaharan, 2016).

## 3. Related Works

### 3.1. Offline Reinforcement Learning

Offline RL is the problem of policy optimization with previously collected data only, which is well-known for suffering from the value overestimation problem for out-of-distribution states and actions. Previous methods for solving this issue include explicit behavioral regularization (Wu et al., 2019; Fujimoto & Gu, 2021; Brandfonbrener et al., 2021), pessimistic value estimation (Kumar et al., 2020; Lyu et al., 2022; Kostrikov et al., 2022), out-of-distribution detection (Yu et al., 2020; Kidambi et al., 2020; An et al., 2021) and dual RL (Lee et al., 2021; Sikchi et al., 2024).

### 3.2. Offline-to-Online RL

Besides the advancements only using offline data, recent works starts to allow the offline-pretrained RL agents to be finetuned through online interactions, referred to as offline-to-online RL. In this regime, we fine-tune the policy with additional online rollouts, and this process usually suffers from a catastrophic degraded performance at initial online training stage due to the distribution shift of training sam-

ples. Prior research has studied online fine-tuning with offline data or pre-trained policies, including Off2OnRL (Lee et al., 2022), Hybrid Q learning (Song et al., 2023b), Cal-QL (Nakamoto et al., 2023), RLPD (Ball et al., 2023) and ACA (Yu & Zhang, 2023). Our method, MoMa QL, mainly focuses on offline RL setting, but we will also show it can also be directly fine-tuned with online rollouts.

### 3.3. Score-based and Flow-based Decision Making

Previous works have applied various iterative score-based models to a series of RL tasks. Diffusion QL (Wang et al., 2023) initiates the usage of DDPM (Ho et al., 2020) as policy representation in a Q-Learning+BC framework. Implicit Diffusion Q-learning (IDQL) (Hansen-Estruch et al., 2023) improves IQL via integrating diffusion policy. Diffusion policies (Chi et al., 2025) applies DDIM model for both state-based and vision-based imitation learning tasks in Robotics domain. Consistency-AC (Ding & Jin, 2024) utilizes CM to accelerate the action sampling. Methods such as QGPO (Lu et al., 2023), EDP (Kang et al., 2023) and QIPO (Zhang et al., 2025b) model the offline RL objective as an energy-guided diffusion process, which can be viewed as a variant of AWR (Peng et al., 2019; Nair et al., 2021). Flow matching is also widely used in offline RL. GFlower (Zheng et al., 2023) apply flow matching models to policy optimization. FlowPolicy (Zhang et al., 2025a) employed Consistency Flow Matching for robot manipulation tasks. FQL (Park et al., 2025) introduced a distilled neural network for a flow-based policy, which accelerates the inference. In our experiments, we compare our method against a representative subset of these models to demonstrate its superior empirical performance and inference efficiency.

## 4. Moment Matching Q-Learning

We now introduce our method, *Moment Maching Q-Learning* (MoMa QL), for effective decision-making tasks. The cornerstone of the whole algorithm is to learn an implicit action sampler which can harness the expressive power of score and flow based models and expedite the training and inference process.

### 4.1. General Objective

For the generative policies, the model is targeted to learn an implicit conditional sampler for $p^\theta(\mathbf{x}|\boldsymbol{\varepsilon}, \mathbf{s})$, where $\mathbf{x} \sim q(\mathbf{x})$, $\boldsymbol{\varepsilon} \sim p(\boldsymbol{\varepsilon})$ and $\mathbf{s}$ denotes the state of the agent which serves as the guidance for the reversed process. In the BRAC framework, the loss function in Eq. (2) can be rewritten as:

$$\mathcal{L}_\pi(\theta) = \mathbb{E}_{\mathbf{s}, \mathbf{a} \sim \mathcal{D}, \mathbf{a}^\pi \sim \pi_\theta} [\underbrace{-\eta Q_\phi(\mathbf{s}, \mathbf{a}^\pi)}_{\text{Q loss}}] + \underbrace{\mathcal{L}_D(\theta)}_{\text{BC loss}}, \quad (4)$$

where $D$ is usually designed as a sample-based metric to calculate the divergence between the marginal $p$ and $q$ measure.

We utilize MMD loss due to its great optimization stability and superior statistical property, and thus match higher order moment of the distribution. Details related to MMD will be further introduced in Appendix A.1. For the simplicity, we will ignore the guidance **s** in further derivation.

## 4.2. Marginal Interpolants

Assuming a time-augmented interpolation $\mathbf{x}_t \sim q_t(\mathbf{x}_t|\mathbf{x}, \varepsilon)$ between original data distribution $q(\mathbf{x})$ and prior distribution $p(\varepsilon)$, we can formulate the marginal interpolating distribution as:

$$q_t(\mathbf{x}_t) = \iint q_t(\mathbf{x}_t|\mathbf{x}, \varepsilon)q(\mathbf{x})p(\varepsilon)\mathrm{d}\mathbf{x}\mathrm{d}\varepsilon. \quad (5)$$

Suppose the conditional distribution of $\mathbf{x}_s|\mathbf{x}_t, \mathbf{x}$ follows as:

$$q_{s|t}(\mathbf{x}_s|\mathbf{x}, \mathbf{x}_t) = \mathcal{N}\left(I_{s|t}(\mathbf{x}, \mathbf{x}_t), \gamma_{s|t}^2 I\right) \quad (6)$$

with constraints $I_{t|t}(\mathbf{x}, \mathbf{x}_t) = \mathbf{x}_t, I_{0|t}(\mathbf{x}, \mathbf{x}_t) = \mathbf{x}, \gamma_{t|t} = \gamma_{0|t} = 0$. This is a natural extrapolation of conditional interpolation $q_t(\mathbf{x}_t|\mathbf{x}, \varepsilon)$. Conceptually, it extends the traditional framework by dynamically shifting the prior from endpoint 1 to an arbitrary time $t$. We also adopt the notion of *marginal-preserving* (Zhou et al., 2025) as follows:

**Definition 4.1** (Marginal-Preserving Interpolants). A generalized interpolant $\mathbf{x}_s$ is *marginal-preserving* if for all $t \in [0, 1]$ and for all $s \in [0, t]$, the following equality holds:

$$q_s(\mathbf{x}_s) = \iint q_{s|t}(\mathbf{x}_s|\mathbf{x}, \mathbf{x}_t)q_t(\mathbf{x}|\mathbf{x}_t)q_t(\mathbf{x}_t)\mathrm{d}\mathbf{x}_t\mathrm{d}\mathbf{x}, \quad (7)$$

where

$$q_t(\mathbf{x}|\mathbf{x}_t) = \int \frac{q_t(\mathbf{x}_t|\mathbf{x}, \varepsilon)q(\mathbf{x})p(\varepsilon)}{q_t(\mathbf{x}_t)}\mathrm{d}\varepsilon. \quad (8)$$

This constraint can provide strong theoretical guarantee for the existence and convergence of the minimizer, as further elucidated in Appendix A.2. Similarly, for $\forall t \in [0, 1]$ and $\forall s \in [0, t]$, we can thus define the marginal model on $p$ measure as:

$$p_{s|t}^\theta(\mathbf{x}_s) = \iint q_{s|t}(\mathbf{x}_s|\mathbf{x}, \mathbf{x}_t)p_{s|t}^\theta(\mathbf{x}|\mathbf{x}_t)q_t(\mathbf{x}_t)\mathrm{d}\mathbf{x}_t\mathrm{d}\mathbf{x} \quad (9)$$

where the interpolate is marginal preserving and $p_{s|t}^\theta(\mathbf{x}|\mathbf{x}_t)$ serves as the implicit one-step sampler we intend to attain. So, our main target is to build a model which can help us to minimize the gap between $p_{s|t}^\theta(\mathbf{x}_s)$ and $q_s(\mathbf{x}_s)$.

To produce a clean sample $\mathbf{x}$ given $\mathbf{x}_t \sim q(\mathbf{x}_t)$ via an intermediate $s$, we first sample a denoised $\tilde{\mathbf{x}} \sim p_{s|t}^\theta(\mathbf{x}|\mathbf{x}_t)$ and then sample $\tilde{\mathbf{x}}_s \sim q_{s|t}(\mathbf{x}_s|\mathbf{x}, \mathbf{x}_t)$ and then (3) sample $\mathbf{x} \sim p_{0|s}^\theta(\mathbf{x}|\tilde{\mathbf{x}}_s)$. Similar to DDIM (**?**), this process can be inductive, and thus we can also obtain a multi-step sampler for $p^\theta$ from $t = 1$ to $t = 0$.

## 4.3. Actor Loss Formulation

Directly matching the MMD loss between $q_s(\mathbf{x}_s)$ and $p_{s|t}^\theta(\mathbf{x}|\mathbf{x}_t)$ is not a good idea: there can be great discrepancy between $q_t(\mathbf{x}_t)$ and $q_s(\mathbf{x}_s)$ when $t$ is far from $s$, so $p_{s|t}^\theta(\mathbf{x}_s)$ can be difficult to estimate. We adopt a training paradigm inspired by consistency training (Song et al., 2023a) to address this challenge. Specifically, we introduce an intermediate timestep $r$ such that $s < r < t$, and enforce a self-consistency constraint by aligning the denoising distributions $p_{s|r}^\theta(\mathbf{x}_s)$ and $p_{s|t}^\theta(\mathbf{x}_s)$:

$$\mathcal{L}_D(\theta) = \mathbb{E}_{s,r,t}\left[\mathrm{MMD}^2\left(p_{s|r}^{\theta^-}(\mathbf{x}_s), p_{s|t}^\theta(\mathbf{x}_s)\right)\right] \quad (10)$$

We can utilize a neural network $G_\theta(\mathbf{x}, s, t)$ to attain the denoised sample $\tilde{\mathbf{x}}$ and thus define $p_{s|t}^\theta(\mathbf{x}|\mathbf{x}_t) = \delta(\mathbf{x} - G_\theta(\mathbf{x}_t, s, t))$. With the denoised sample $\tilde{\mathbf{x}}$, we choose to utilize DDIM interpolants, denoted as $f_{s,t}^\theta(\mathbf{x}_t)$ and $f_{s,r}^\theta(\mathbf{x}_r)$, to obtain a sample from $p_{s|t}^\theta(\mathbf{x}_s)$ and $p_{s|r}^\theta(\mathbf{x}_s)$. Empirically, we choose an unbiased estimator of squared MMD Loss (Gretton et al., 2012) to compute $\mathcal{L}_D$. Further details and theoretical derivation related to actor loss function can be found in Appendix A.3.

## 4.4. Algorithm Implementation

We present a brief MoMa QL algorithm in Algorithm 1 and defer the full version in Appendix B. For parameterized $Q_\phi(\mathbf{s}, \mathbf{a})$, we utilized the double Q-Learning loss (Fujimoto & Gu, 2021) and with batched data $\mathcal{B} \subseteq \mathcal{D}$, which mitigates Q-value overestimation in Eq. (1):

$$\begin{aligned}\mathcal{L}(\phi) = &\mathbb{E}_{(\mathbf{s},\mathbf{a},\mathbf{s}')\sim\mathcal{B},\mathbf{a}'\sim\pi_{\theta^\mathsf{T}}(\cdot|\mathbf{s}')}\Big[\big((r(\mathbf{s},\mathbf{a})+ \\ &\gamma\min_{i\in\{1,2\}}Q_{\phi_i^\mathsf{T}}(\mathbf{s}',\mathbf{a}')) - Q_{\phi_i}(\mathbf{s},\mathbf{a})\big)^2\Big]\end{aligned} \quad (11)$$

The regularized policy $\pi_\theta$ is learned via minimizing the objective function in Eq. (4) via backpropagation. This loss comprises two components: (i) a Q-loss, where actions **a** are sampled inductively according to the procedure in Algorithm 2 and subsequently fed into the critic network $Q_{\phi_i}$; and (ii) a behavior cloning (BC) loss, as formulated in Eq. (10). We update the actor and critic networks using an exponential moving average (EMA) with a smoothing coefficient $\alpha$.

# 5. Experiments

This section empirically evaluates our proposed RL algorithm, MoMa QL. We conduct comprehensive experiments on standard offline RL and offline-to-online RL benchmarks. We also conducted experiments to demonstrate the effectiveness and stability of our method.

---

**Algorithm 1** Moment Matching Q-Learning

**Input** offline dataset $\mathcal{D}$
Initialize policy network $\pi_\theta$, critic networks $Q_{\phi_1}, Q_{\phi_2}$
Initialize target network parameters: $\theta^\mathsf{T} \leftarrow \theta, \phi_1^\mathsf{T} \leftarrow \phi_1, \phi_2^\mathsf{T} \leftarrow \phi_2$
**while** not converge **do**
    Sample batch $\mathcal{B} = \{(\mathbf{s}, \mathbf{a}, r, \mathbf{s}')\} \subseteq \mathcal{D}$;
    // Q-value Update
    Update $Q_{\phi_1}, Q_{\phi_2}$ via Eq. (11);
    // Policy Update
    Update policy $\pi_\theta$ via Eq. (4);
    // Target Update
    Update target: $\theta^\mathsf{T} \leftarrow \alpha\theta^\mathsf{T} + (1-\alpha)\theta, \phi_i^\mathsf{T} \leftarrow \alpha\phi_i^\mathsf{T} + (1-\alpha)\phi_i, i \in \{1,2\}$;
**end while**
**return** $\pi_\theta, Q_{\phi_1}, Q_{\phi_2}$

---

**Algorithm 2** Action Inference

**Input** network $f^\theta$, time schedule $\{t_i\}_{i=0}^N$
Initialize $\mathbf{a}_N^\pi \sim \mathcal{N}(0, \sigma_d^2 I)$
**for** $i = N, \dots, 1$ **do**
    $\mathbf{a}_{t_{i-1}}^\pi \leftarrow f_{t_{i-1}, t_i}^\theta(\mathbf{a}_{t_i}^\pi)$
**end for**
**return** $\mathbf{a}^\pi := \mathbf{a}_{t_0}^\pi$

---

### 5.1. Experiment Setup

#### 5.1.1. BENCHMARKS

We primarily evaluate the expressiveness and computational efficiency of our proposed algorithm and the responding algorithms on three task suites (Gym, Adroit and Kitchen) in D4RL benchmarks (Fu et al., 2021). D4RL benchmarks include a diverse set of offline RL tasks spanning locomotion, manipulation, and Gym Mujoco environments. Tasks in D4RL are formulated with fixed offline datasets and standardized evaluation protocols.

#### 5.1.2. BASELINE METHODS

Our experiments evaluate three distinct learning paradigms: behavior cloning (BC), offline reinforcement learning (RL), and offline-to-online RL. We select a range of established and state-of-the-art algorithms as baselines, drawing from both classical approaches and recent advances in generative modeling for RL.

**Behavior Cloning (BC).** We compare our method with vanilla Behavior Cloning (BC) parameterized by Gaussian policy, Consistency-BC (Ding & Jin, 2024), and Diffusion-BC (Wang et al., 2023). Additionally, we include classic BC baselines reported in previous works: Diffuser (Janner et al., 2022), MoRel (Kidambi et al., 2020), Onestep RL

(Brandfonbrener et al., 2021), TD3+BC (Fujimoto & Gu, 2021) and Decision Transformer(DT)(Chen et al., 2021).

**Offline RL.** For offline RL, we compare methods across different algorithmic families. In Gym locomotion tasks, baselines include Conservative Q-Learning (CQL) (Kumar et al., 2020), Implicit Q-Learning (IQL) (Kostrikov et al., 2022), Extreme Q-learning ($\mathcal{X}$-QL)(Garg et al., 2023), Actor-Restricted Q-learning (ARQ)(Goo & Niekum, 2022), IDQL-A(Hansen-Estruch et al., 2023), Diffusion-QL (Wang et al., 2023), and Consistency-AC (CAC) (Ding & Jin, 2024). For Adroit manipulation tasks, we consider BC, IQL(Kostrikov et al., 2022), ReBRAC (Tarasov et al., 2023), Implicit Diffusion Q-Learning (IDQL) (Hansen-Estruch et al., 2023), SRPO (Chen et al., 2024), CAC(Ding & Jin, 2024), Flow advantage-weighted actor-critic (FAWAC) (Nair et al., 2021), Flow behavior-regularized actor-critic (FBRAC)(Wang et al., 2023), Implicit Flow Q-Learning (IFQL)(Park et al., 2025), and Flow Q-Learning (FQL) (Park et al., 2025). For Kitchen tasks, baselines include CQL, IQL, $\mathcal{X}$-QL, ARQ, Diffusion-QL, and Consistency-AC.

**Offline-to-Online RL.** In the offline-to-online setting, we initialize the policy using offline data and fine-tune it with online interactions. We compare MoMa QL against IQL, Cal-QL (Nakamoto et al., 2023), RLPD (Ball et al., 2023), Diffusion-QL, and Consistency-AC. We also include comparisons with Soft Actor-Critic (SAC) (Haarnoja et al., 2018), Advantage-Weighted Actor-Critic (AWAC) (Nair et al., 2021), and Actor-Critic Alignment (ACA) (Yu & Zhang, 2023) in detailed evaluations.

For each method, We adopted the same results from the previous works(Ding & Jin, 2024; Park et al., 2025). We do not use the same parameters for all the methods.

### 5.2. Main Results For Behavior Cloning

We evaluate our method, MoMa QL, against various baseline algorithms in the behavior cloning (BC) setting. Table 1 summarizes the performance comparisons on the D4RL Gym tasks. The hyperparameters for training can be found in Appendix C.1.

The quantitative results indicate that our method, MoMa QL, achieves the highest overall average score of 89.8, surpassing all baseline methods. Compared to vanilla BC (51.9) and Diffusion-BC (76.3), our approach achieves relative improvements of $1.73\times$ and $1.18\times$, respectively. This suggests that the moment-matching objective effectively captures the underlying distribution of the dataset, providing a more robust policy representation than standard supervised learning or diffusion-based counterparts.

Notably, MoMa QL excels in tasks with medium and medium-replay datasets. For instance, on `walker2d-mr` and `halfcheetah-m`, our method surpasses the strongest

*Table 1.* Performance comparison on D4RL Gym tasks under the Behavior Cloning setting. We report the normalized average scores over five random seeds with standard deviations reported. The best result for each task is highlighted in bold.

| Task Category | BC | Consistency-BC | Diffusion-BC | Diffuser | MoRel | Onestep RL | TD3+BC | DT | Ours |
|---|---|---|---|---|---|---|---|---|---|
| halfcheetah-m | 42.6 | 31.0±0.4 | 45.4±1.8 | 44.2 | 42.1 | 48.4 | 48.3 | 42.6 | **69.9± 1.2** |
| hopper-m | 52.9 | 71.7±8.0 | 65.3±5.8 | 58.5 | 95.4 | 59.6 | 59.3 | 67.6 | **104.2±4.6** |
| walker2d-m | 75.3 | 83.1±0.3 | 81.2±1.7 | 79.7 | 77.8 | 81.8 | 83.7 | 74.0 | **87.5±0.7** |
| halfcheetah-mr | 36.6 | 34.4±5.3 | 41.7±0.4 | 42.2 | 40.2 | 38.1 | 44.6 | 36.6 | **58.3±2.7** |
| hopper-mr | 18.1 | 99.7±0.5 | 67.9±28.1 | 96.8 | 93.6 | 97.5 | 60.9 | 82.7 | **105.8±1.3** |
| walker2d-mr | 26.0 | 73.3±5.7 | 77.5±4.7 | 61.2 | 49.8 | 49.5 | 81.8 | 66.6 | **103.0±3.3** |
| halfcheetah-me | 55.2 | 32.7±1.2 | 90.8±1.1 | 79.8 | 53.3 | 93.4 | 90.7 | 86.8 | **106.9±2.3** |
| hopper-me | 52.5 | 90.6±9.3 | **107.6±4.3** | 107.2 | 108.7 | 103.3 | 98.0 | 107.6 | 67.9±12.5 |
| walker2d-me | 107.5 | 110.4±0.7 | 108.9±0.6 | 108.4 | 95.6 | **113.0** | 110.1 | 108.1 | 104.6±1.3 |
| **Average** | 51.9 | 69.7 | 76.3 | 75.3 | 72.9 | 76.1 | 75.3 | 74.7 | **89.8** |

baselines by factors of $1.26\times$ and $1.44\times$, respectively. This highlights the effectiveness of our method in handling suboptimal and highly stochastic data. While the performance on `hopper-me` is lower than some baselines, likely due to the specific characteristics of that expert dataset, the consistent superiority across the majority of tasks confirms the strong generalization capability of our proposed framework.

### 5.3. Results For Offline RL

We evaluate MoMa QL on standard D4RL benchmarks across three task suites: Gym locomotion tasks, Adroit manipulation tasks, and Kitchen tasks. Following key baselines, we compare MoMa QL against a variety of methods representing distinct policy classes. Table 2 presents the aggregated results across dataset variants. Detailed per-dataset results can be found in Table 5 in Appendix C.2.

**Gym Locomotion Tasks.** In the standard D4RL Gym locomotion suite, MoMa QL achieves a dominant average score of **95.5**, surpassing the strongest baseline, Diffusion-QL (87.9), by a factor of approximately $1.09\times$. Specifically, on HalfCheetah and Walker2D, our method outperforms the nearest competitors by margins of $14\%$ and $8.5\%$, respectively. Although slightly trailing Diffusion-QL on Hopper (99.9 vs. 101.0), MoMa QL demonstrates superior consistency across tasks, effectively balancing distribution matching with reward maximization.

**Adroit Manipulation Tasks.** For the challenging high-dimensional Adroit tasks, MoMa QL achieves an average score of **56.7**, which is highly competitive with the state-of-the-art method ReBRAC (55.4). Notably, on the `door` task, our method attains the highest score of **38.5**, outperforming all baselines. While ReBRAC shows advantages in expert datasets, MoMa QL demonstrates robustness across diverse dataset qualities, particularly excelling in scenarios where capturing multi-modal distributions is critical.

**Kitchen Tasks.** In the long-horizon Kitchen environments, MoMa QL achieves a strong average score of **73.1**, out-

performing Diffusion-QL (69.0) and $\mathcal{X}$-QL (72.9). This highlights the advantage of our sample-efficient few-step generation for sequential control tasks requiring long-term planning.

**Overall Performance.** Across all benchmarks, MoMa QL demonstrates consistent state-of-the-art or competitive performance. The significant improvements in Gym tasks ($1.09\times$ over Diffusion-QL) and robust results in Adroit and Kitchen domains validate the effectiveness of our MMD-based moment matching framework.

### 5.4. Offline-to-Online Fine-tuning

We further investigate the capability of MoMa QL in the offline-to-online setting. In this regime, we initialize the policy with the pre-trained offline model (trained for 100K steps) and then fine-tune it with online interactions for an additional 400K steps. We use the same hyperparameters as in the offline setting, which are detailed in Appendix C.1.

Table 3 highlights the performance improvement of MoMa QL after online fine-tuning. Our method achieves significant gains, particularly on the challenging medium-replay datasets, where the offline initialization provides a strong starting point for further effective learning. For instance, on `halfcheetah-medium-replay`, performance improves from 63.3 to 80.9 ($\approx 28\%$ increase).

Figure 1 visualized the improvement of MoMa QL after online finetuning. More detailed comparisons with baselines are provided in Appendix C.3.

### 5.5. Ablation Studies

We conduct comprehensive ablation studies to demonstrate that MoMa QL is highly robust to hyperparameter variations. We examine six parameters: $\eta$, $N$, $a$, $b$, $p_{\text{mean}}$, and $p_{\text{std}}$. Detailed results are provided in Appendix C.4.

First, we analyze the impact of the Q-learning weight $\eta$ and

*Table 2.* Aggregated offline RL results on D4RL benchmarks. We report normalized scores averaged across dataset variants (e.g., medium, medium-replay, medium-expert). We report the normalized average scores over five random seeds with standard deviations reported. Bold values indicate the best performance in each row.

| **Gym Tasks** | CQL | IQL | $\mathcal{X}$-QL | ARQ | IDQL-A | Diffusion-QL | Consistency-AC | MoMa QL (Ours) |
|---|---|---|---|---|---|---|---|---|
| halfcheetah (3 tasks) | 60.4 | 59.4 | 62.6 | 59.3 | 64.3 | 65.2 | 70.7 | **80.9** |
| hopper (3 tasks) | 86.3 | 84.2 | 95.4 | 84.0 | 88.7 | **101.0** | 93.6 | 99.9 |
| walker2d (3 tasks) | 86.2 | 87.3 | 93.0 | 85.3 | 93.4 | 97.5 | 91.0 | **105.8** |
| Average | 77.6 | 77.0 | 83.7 | 76.2 | 82.1 | 87.9 | 85.1 | **95.5** |

| **Adroit Tasks** | BC | IQL | ReBRAC | IDQL | SRPO | CAC | FAWAC | FBRAC | IFQL | FQL | MoMa QL (Ours) |
|---|---|---|---|---|---|---|---|---|---|---|---|
| pen (3 tasks) | 77.7 | 96.3 | **119.3** | 93.3 | 88.0 | 74.3 | 82.3 | 87.7 | 96.7 | 89.7 | 108.3 |
| door (3 tasks) | 35.7 | 37.7 | 35.3 | 37.0 | 36.0 | 34.7 | 35.0 | 36.0 | 37.7 | 35.3 | **38.5** |
| hammer (3 tasks) | 43.7 | 44.3 | 45.3 | 43.0 | 43.3 | 31.7 | 40.3 | 41.0 | 40.7 | 45.7 | **46.2** |
| relocate (3 tasks) | 36.0 | 35.3 | **36.7** | 35.7 | 35.3 | 31.0 | 35.0 | 35.3 | 34.7 | 35.7 | 36.0 |
| Average | 48.3 | 53.4 | 55.4 | 52.3 | 50.7 | 42.9 | 48.2 | 50.0 | 52.4 | 51.6 | **56.7** |

| **Kitchen Tasks** | CQL | IQL | $\mathcal{X}$-QL | ARQ | Diffusion-QL | Consistency-AC | MoMa QL (Ours) |
|---|---|---|---|---|---|---|---|
| kitchen (3 tasks) | 48.2 | 53.3 | 72.9 | 42.0 | 69.0 | 45.3 | **73.1** |

*Table 3.* Performance improvement of MoMa QL in the offline-to-online setting on Gym tasks. We report the score transition from **Offline → Online**. Improvements are substantial across tasks.

| TASK | OFFLINE | ONLINE SCORE | |
|---|---|---|---|
| | SCORE | SCORE | $\Delta$ |
| HALFCHEETAH-M | 72.6 | **83.1** | +10.5 |
| HOPPER-M | 104.2 | **104.3** | +0.1 |
| WALKER2D-M | 95.6 | **99.1** | +3.5 |
| HALFCHEETAH-MR | 63.3 | **80.9** | +17.6 |
| HOPPER-MR | 106.5 | **113.5** | +7.0 |
| WALKER2D-MR | 104.3 | **117.9** | +13.6 |
| HALFCHEETAH-ME | 106.9 | **107.1** | +0.2 |
| HOPPER-ME | 88.9 | **89.1** | +0.2 |
| WALKER2D-ME | 117.5 | **118.0** | +0.5 |

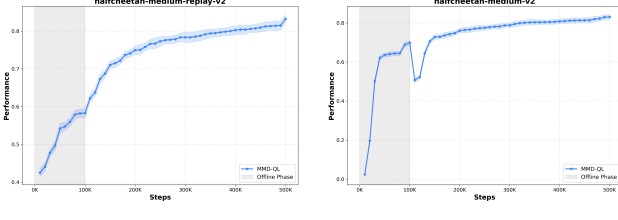

*(a)* offline to online on halfcheetah-medium-replay

*(b)* offline to online on halfcheetah-medium

*Figure 1.* Learning curves of MoMa QL during online fine-tuning on `halfcheetah-medium-replay` and `halfcheetah-medium`. The offline initialization allows for rapid adaptation and consistent improvement.

sampling steps $N$. As shown in Figure 2(a), performance remains stable and high for $\eta \in [0.5, 1.0]$. This indicates that the method is not sensitive to precise tuning of the balance parameter, effectively learning across a broad range of weights. Regarding the number of sampling steps, Figure 2(b) demonstrates that performance saturates at $N = 2$. MoMa QL is thus computationally efficient, working well with as few as 2-4 steps without performance degradation at higher steps.

Furthermore, MoMa QL performs consistently well across variations in MMD kernel parameters $(a, b)$ and noise schedule parameters $(p_{\text{mean}}, p_{\text{std}})$. Figure 3(a, b) shows that the method is insensitive to kernel parameters $a, b \in [1, 8]$, requiring no specific tuning. Similarly, Figure 3(c, d) indicates that performance is largely invariant to the noise schedule parameters. This overall robustness ensures that MoMa QL

is easy to deploy without complex hyperparameter search.

### 5.6. Computational Cost Analysis

We evaluate the computational efficiency of MoMa QL by measuring the training time per 1,000 gradient steps across different tasks. We ensure a fair comparison by using identical batch sizes across all methods. Figure 4 compares the training efficiency of our method against diffusion-based (Diffusion-BC) and consistency-based (Consistency-BC) baselines. Baseline results are normalized from literature values, assuming 1 epoch = 500 steps vs. our 1,000 steps.

The results demonstrate that MoMa QL achieves superior computational efficiency. On average, our method is approximately **6×** faster than Diffusion-BC and **1.5×** faster than Consistency-AC on complex tasks like Adroit and Kitchen. Notably, MoMa QL maintains a constant train-

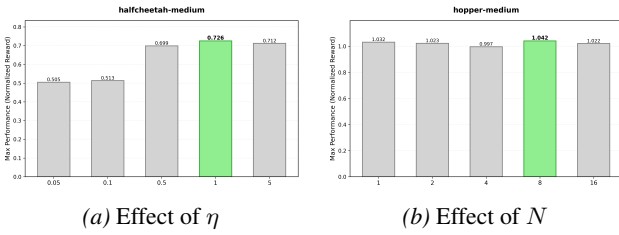

*(a)* Effect of $\eta$    *(b)* Effect of $N$

*Figure 2.* Robustness analysis for $\eta$ and $N$. (a) Performance is consistent across $\eta \in [0.5, 5.0]$. (b) High performance is maintained for $N \geq 2$.

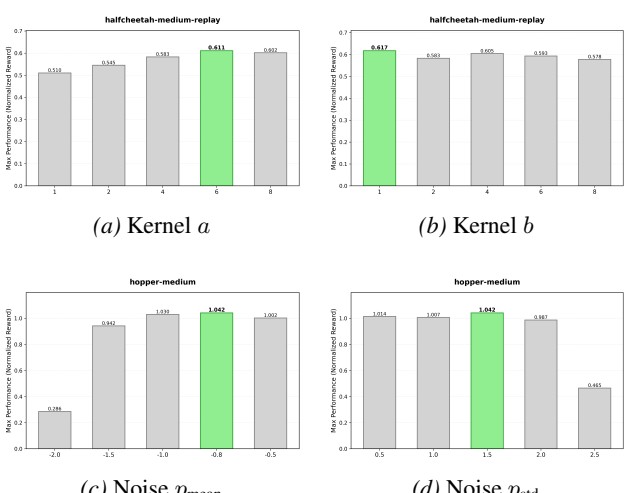

*(a)* Kernel $a$    *(b)* Kernel $b$

*(c)* Noise $p_{\text{mean}}$    *(d)* Noise $p_{\text{std}}$

*Figure 3.* Robustness analysis. Performance is stable across variations in kernel parameters $(a, b)$ and noise parameters $(p_{\text{mean}}, p_{\text{std}})$.

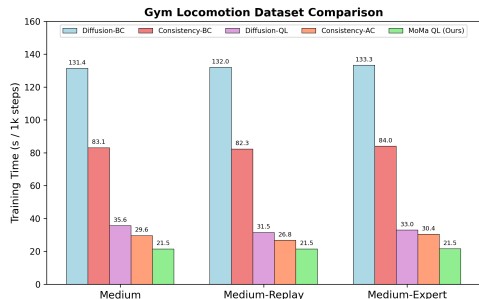

*Figure 4.* Training time comparison (seconds per 1,000 steps) across D4RL domains. MoMa QL demonstrates superior scalability and efficiency, particularly in high-dimensional Adroit and Kitchen environments.

are provided in Appendix C.5.2.

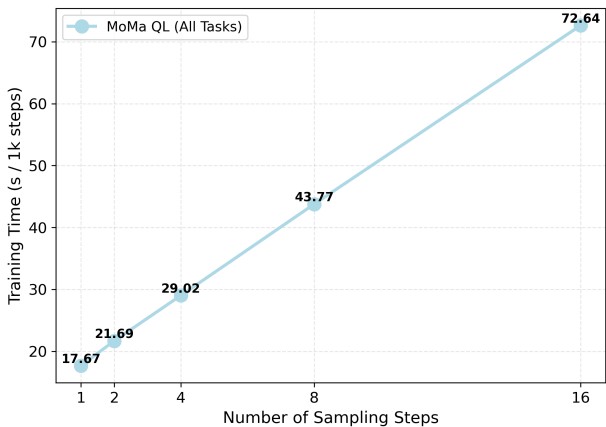

*Figure 5.* Training time (seconds per 1,000 steps) with respect to the sampling steps for MoMa QL. We report average training time for Gym, Adroit, and Kitchen environments.

Detailed per-task training and inference time comparisons are provided in Appendix C.5.

ing cost ($\approx 21 - 23$ seconds per 1k steps) regardless of task dimensionality, whereas baselines exhibit significant slowdowns on high-dimensional tasks.

**Impact of Sampling Steps.** We further analyze how the number of sampling steps affects training efficiency. Figure 5 shows the relationship between the number of sampling steps (1, 2, 4, 8, 16) and training time per 1,000 gradient steps across Gym, Adroit, and Kitchen domains.

The results reveal several key insights: (1) Training time scales approximately linearly with the number of sampling steps, increasing from ~17-18s for single-step sampling to ~71-73s for 16-step sampling across all domains. (2) MoMa QL maintains consistent efficiency regardless of task complexity—the training cost remains nearly identical across Gym locomotion ($17.70 \rightarrow 72.56$s), Adroit manipulation ($17.64 \rightarrow 73.12$s), and Kitchen ($17.71 \rightarrow 70.94$s) tasks, demonstrating excellent scalability. (3) Even with 16 sampling steps, MoMa QL remains faster than Diffusion-BC (~132-189s) and Consistency-BC (~83-130s) baselines. This linear scaling behavior enables practitioners to flexibly trade off between sample quality and computational cost based on task requirements. Detailed per-step timing results

## 6. Conclusion

In this paper, we presented Moment Matching Q-Learning (MoMa QL), an efficient RL framework that integrates MMD regularization with stochastic interpolants. MoMa QL strikes a balance between modeling expressivity and computational efficiency, achieving state-of-the-art results. Empirically, it surpasses Diffusion-QL by $1.09\times$ on Gym tasks and maintains competitive performance on complex Adroit and Kitchen domains. Crucially, our method is approximately and $1.5\times$ faster than Consistency-AC, eliminating the inference bottleneck of generative policies. Beyond efficiency, MoMa QL exhibits strong robustness to hyperparameters and effective offline-to-online fine-tuning capabilities. We hope this simple yet rigorous approach spurs future research into scalable generative policies.

## Acknowledgements

We thank the anonymous reviewers for their helpful comments. This research was supported by WZ's startup funding provided by the School of Data Science and Society at UNC Chapel Hill. This research supported by the NVIDIA Academic Grant Program with the cloud GPU credits and infrastructure.

## Impact Statement

This paper presents work whose goal is to advance the field of Reinforcement Learning and Robotics. There are many potential societal consequences of our work, none which we feel must be specifically highlighted here.

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

# A. Theorems and Derivations

## A.1. Maximum Mean Discrepancy

To facilitate a further discussion of MMD, it is necessary to first introduce several key mathematical definitions.

**Definition A.1** (Integral Probability Metric). A metric $D(\cdot, \cdot)$ between two probability measures $\mu, \nu$ is called an *integral probability metric* (IPM) if it satisfies the properties of a metric and can be written in the form

$$D(\mu, \nu) = \sup_{f \in \mathcal{F}} \left| \int f \, \mathrm{d}\mu - \int f \, \mathrm{d}\nu \right|. \tag{12}$$

where $\mathcal{F}$ is a function class.

For instance, 1-Wasserstein distance $W_1$ is an IPM equipped with function class $\mathcal{F} = \mathrm{Lip}_1 := \{f : \mathcal{X} \to \mathbb{R} \mid |f(x) - f(y)| \leq d(x, y), \forall x, y \in \mathcal{X}\}$, which is exactly the loss function in WGAN (Arjovsky et al., 2017). We mentioned *Reproducing Kernel Hilbert Space* (RKHS) in section 2.4, and now we give the formal definition of RKHS as:

**Definition A.2** (Reproducing Kernel Hilbert Space). Let $\mathcal{H}$ be a Hilbert space of functions defined on a set $\mathcal{X}$, $\mathcal{H} \subseteq \{f : \mathcal{X} \to \mathbb{F}\}$ ($\mathbb{F} = \mathbb{R}$ or $\mathbb{C}$). If $\forall x \in X$, the evaluation functional $L_x : \mathcal{H} \to \mathbb{F}$, $L_x(f) = f(x)$ is a continuous linear functional, then $\mathcal{H}$ is called a *Reproducing Kernel Hilbert Space* (RKHS).

Since $L_x$ is continuous and linear, we can apply *Riesz Representation Theorem* presented as follows, and thus prove the existence of a kernel function $K(\cdot, \cdot)$, $K_x := K(x, \cdot) \in \mathcal{H}$ which satisfies the $f(x) = \langle f(\cdot), K(\cdot, x) \rangle_{\mathcal{H}}$ (referred as "reproducing property").

**Theorem A.3** (Riesz Representation Theorem). *Let $\mathcal{H}$ be a Hilbert space over $\mathbb{F}$, where $\mathbb{F} = \mathbb{R}$ or $\mathbb{C}$. Assume that the inner product $\langle \cdot, \cdot \rangle_{\mathcal{H}}$ is linear in the first argument and conjugate-linear in the second. For any continuous linear functional $\varphi : \mathcal{H} \to \mathbb{F}$, there exists a unique element $y \in \mathcal{H}$ such that*

$$\varphi(h) = \langle h, y \rangle_{\mathcal{H}}, \quad \forall h \in \mathcal{H}. \tag{13}$$

*Proof.* If $\varphi = 0$, the statement holds trivially by choosing $y = 0$. Hence, we assume $\varphi \neq 0$.

**Existence.** Let

$$M := \ker(\varphi) = \{h \in \mathcal{H} : \varphi(h) = 0\}.$$

Since $\varphi$ is continuous and linear, $M$ is a closed proper subspace of $\mathcal{H}$. By the Projection Theorem, we have the orthogonal decomposition

$$\mathcal{H} = M \oplus M^{\perp}.$$

Because $\varphi \neq 0$, the orthogonal complement $M^{\perp}$ is nontrivial. Choose a nonzero vector $z \in M^{\perp}$ and normalize it so that $\|z\|_{\mathcal{H}} = 1$. Note that $\varphi(z) \neq 0$, since otherwise $z \in \ker(\varphi) = M$, contradicting $z \in M^{\perp} \setminus \{0\}$.

For any $h \in \mathcal{H}$, define

$$u := h - \frac{\varphi(h)}{\varphi(z)} z.$$

Then

$$\varphi(u) = \varphi(h) - \frac{\varphi(h)}{\varphi(z)} \varphi(z) = 0,$$

which implies $u \in M$. Since $z \in M^{\perp}$, we have $\langle u, z \rangle_{\mathcal{H}} = 0$, and thus

$$\left\langle h - \frac{\varphi(h)}{\varphi(z)} z, z \right\rangle_{\mathcal{H}} = 0.$$

Expanding the inner product and using $\langle z, z \rangle_{\mathcal{H}} = 1$, we obtain

$$\langle h, z \rangle_{\mathcal{H}} - \frac{\varphi(h)}{\varphi(z)} = 0.$$

Hence,

$$\varphi(h) = \varphi(z)\langle h, z \rangle_{\mathcal{H}} = \langle h, \overline{\varphi(z)}\, z \rangle_{\mathcal{H}},$$

where the complex conjugation appears due to conjugate-linearity in the second argument. Defining

$$y := \overline{\varphi(z)}\, z \in \mathcal{H},$$

we conclude that (13) holds for all $h \in \mathcal{H}$.

**Uniqueness.** Suppose there exist $y_1, y_2 \in \mathcal{H}$ such that

$$\varphi(h) = \langle h, y_1 \rangle_{\mathcal{H}} = \langle h, y_2 \rangle_{\mathcal{H}}, \quad \forall h \in \mathcal{H}.$$

Then

$$\langle h, y_1 - y_2 \rangle_{\mathcal{H}} = 0, \quad \forall h \in \mathcal{H}.$$

Taking $h = y_1 - y_2$ yields

$$\|y_1 - y_2\|_{\mathcal{H}}^2 = 0,$$

which implies $y_1 = y_2$. This proves uniqueness. $\qquad\square$

We can also prove the finiteness of MMD with this useful proposition.

**Proposition A.4.** *Let $\mathcal{H}$ be an RKHS with kernel $k$. The Maximum Mean Discrepancy (MMD) between two probability measures $\mu$ and $\nu$ on $\mathbb{R}^d$ can be equivalently defined as:*

$$MMD(\mu, \nu) = \left\| \int_{\mathbb{R}^d} k(x, \cdot)\mu(dx) - \int_{\mathbb{R}^d} k(x, \cdot)\nu(dx) \right\|_{\mathcal{H}}. \tag{14}$$

*Proof.* For any function $f \in \mathcal{H}$, by the reproducing property $f(x) = \langle k(x, \cdot), f \rangle_{\mathcal{H}}$, it holds that:

$$\int f(x)(\mu - \nu)(dx) = \int \langle k(x, \cdot), f \rangle_{\mathcal{H}}(\mu - \nu)(dx) \tag{15}$$

$$= \left\langle \int k(x, \cdot)(\mu - \nu)(dx), f \right\rangle_{\mathcal{H}}. \tag{16}$$

The claim then follows directly from the application of the Cauchy–Schwarz inequality. $\qquad\square$

**Corollary A.5.** *As a corollary of Proposition A.4, the squared MMD is given by:*

$$MMD^2(\mu, \nu) = \iint k(x, y)\mu(dx)\mu(dy) + \iint k(x, y)\nu(dx)\nu(dy)$$

$$- 2 \iint k(x, y)\mu(dx)\nu(dy). \tag{17}$$

Furthermore, the MMD is bounded if the kernel $k$ is bounded:

$$\mathrm{MMD}(\mu, \nu) \leq \int \|k(x, \cdot)\|_{\mathcal{H}}\mu(dx) + \int \|k(x, \cdot)\|_{\mathcal{H}}\nu(dx) \tag{18}$$

$$\leq 2 \sup_{x \in \mathbb{R}^d} \sqrt{k(x, x)} < \infty, \tag{19}$$

where we utilize the fact that $\|k(x, \cdot)\|_{\mathcal{H}} = \sqrt{\langle k(x, \cdot), k(x, \cdot) \rangle_{\mathcal{H}}} = \sqrt{k(x, x)}$. Unlike Wasserstein distances, MMD does not suffer from the curse of dimensionality, and thus is a powerful metric for matching the gap between high-dimensional distributions.

## A.2. Existence of Divergence Minimizer

In this section, we intend to prove the existence of the optimal $p_{s|t}^{\theta^*}$, and we can further prove that this optimal solution can be learnt inductively in the form of Eq. (10).

**Lemma A.6.** *Assuming marginal-preserving interpolant and metric $D(\cdot, \cdot)$, there exists an optimal $\theta^*$ s.t. $p_{s|t}^{\theta^*}(\mathbf{x}|\mathbf{x}_t) = q_t(\mathbf{x}|\mathbf{x}_t)$, and the minimum loss is 0.*

*Proof.* We directly substitute $q_t(\mathbf{x}|\mathbf{x}_t)$ into the objective to check. First, we evaluate the marginal distribution at time $s$:

$$p_{s|t}^{\theta^*}(\mathbf{x}_s) = \iint q_{s|t}(\mathbf{x}_s|\mathbf{x}, \mathbf{x}_t) p_{s|t}^{\theta^*}(\mathbf{x}|\mathbf{x}_t) q_t(\mathbf{x}_t) \, d\mathbf{x}_t d\mathbf{x} \tag{20}$$

$$= \iint q_{s|t}(\mathbf{x}_s|\mathbf{x}, \mathbf{x}_t) q_t(\mathbf{x}|\mathbf{x}_t) q_t(\mathbf{x}_t) \, d\mathbf{x}_t d\mathbf{x} \tag{21}$$

$$= q_s(\mathbf{x}_s) \quad \text{(definition of marginal preservation interpolants)} \tag{22}$$

Consequently, we can have achieve the first stage result as:

$$\mathbb{E}_{s,t} \left[ D(q_s(\mathbf{x}_s), p_{s|t}^{\theta^*}(\mathbf{x}_s)) \right] = \mathbb{E}_{s,t} \left[ D(q_s(\mathbf{x}_s), q_s(\mathbf{x}_s)) \right] = 0 \tag{23}$$

$\square$

In general, the minimizer $q_t(\mathbf{x}|\mathbf{x}_t)$ exists, although it may not be unique. We also should discuss more about the mapping function $0 \leq s \leq r := r(s,t) \leq t \leq 1$. We set $r(s,t) = \max(s, t - \Delta_t)$ where $\Delta_t \geq \epsilon > 0$. In this case, we have can prove the convergence.

**Theorem A.7.** *Assuming $r(s,t)$ as defined above and marginal-preserving interpolants, $\theta^*$ is a minimizer of Eq. (10), then for $\forall n, \forall t \in [0,1], \forall s \in [0,1]$ we have:*

$$\lim_{n \to \infty} \text{MMD}^2(q_s(\mathbf{x}_s), p_{s|t}^{\theta_n^*}(\mathbf{x}_s)) = 0. \tag{24}$$

*Proof.* We can prove this via induction.

For $n = 1$, given $r(s,u) = s$ for $s < u \leq c_0 := \sup\{t : r(s,t) = s\}$, we have:

$$\text{MMD}^2(p_{s|s}^{\theta_0}(\mathbf{x}_s), p_{s|u}^{\theta_1^*}(\mathbf{x}_s)) = \text{MMD}^2(q_s(\mathbf{x}_s), p_{s|u}^{\theta_1^*}(\mathbf{x}_s)) = 0. \tag{25}$$

The first equation holds because we have $q_s(\mathbf{x}_s) = p_{s|s}^{\theta}(\mathbf{x}_s)$, and the second equation holds because of Lemma A.6.

For $n \geq 2$, assume $p_{s|u}^{\theta_{n-1}^*}(\mathbf{x}_s) = q_s(\mathbf{x}_s)$ for $s \leq u \leq c_{n-2}$, where $c_k$ is defined as the jump point for the mapping function $r(s,t)$ when substituting $t$ with $c_{k-1}$ iteratively. We shall inspect the target distribution $p_{s|r(s,u)}^{\theta_{n-1}^*}(\mathbf{x}_s)$ if optimized on $s \leq u \leq c_{n-1}$. On this interval, we know that by inductive assumption minimizing the objective:

$$\mathbb{E}_{s,u}[\text{MMD}^2(p_{s|u}^{\theta_{n-1}^*}(\mathbf{x}_s), p_{s|u}^{\theta_n}(\mathbf{x}_s))]$$

is equivalent to minimizing:

$$\mathbb{E}_{s,u}[\text{MMD}^2(q_s(\mathbf{x}_s), p_{s|u}^{\theta_n}(\mathbf{x}_s))]$$

.

By Lemma A.6, the minimum achieves $p_{s|u}^{\theta_n^*}(\mathbf{x}_s) = q_s(\mathbf{x}_s)$, and we know when $n \to \infty$, $\lim_{n \to \infty} c_n \to 1$, and thus the induction covers the entire $[s, 1]$ interval given each $s$. Therefore, we can prove the original statement. $\square$

## A.3. Empirical Loss Function

For ODE based diffusion model, *DDIM Interpolants* can be formally defined as:

$$\text{DDIM}(\mathbf{x}_t, \mathbf{x}, s, t) = \left(\alpha_s - \frac{\sigma_s}{\sigma_t}\alpha_t\right)\mathbf{x} + \frac{\sigma_s}{\sigma_t}\mathbf{x}_t \tag{26}$$

and sample $\mathbf{x}_s = \text{DDIM}(\mathbf{x}_t, \mathbb{E}_{\mathbf{x}}[\mathbf{x}|\mathbf{x_t}], s, t)$ can be drawn when $\mathbb{E}_{\mathbf{x}}[\mathbf{x}|\mathbf{x_t}]$ is approximated by a network. DDIM can be viewed as an interpolant with $\gamma_{s|t} \equiv 0$ and $I_{s|t} = \text{DDIM}(\mathbf{x}_t, \mathbf{x}, s, t)$, and it is a marginal-preserving interpolants, which means that there is a deterministic sampler which satisfies A.7.

To further justify the marginal-preserving property used in Theorem A.7, we provide a proof sketch showing that DDIM interpolants intrinsically satisfy the marginal-preserving equality.

Consider a DDIM interpolant defined as

$$\mathbf{x}_t = I(\mathbf{x}, \boldsymbol{\epsilon}, t) = a_t\mathbf{x} + b_t\boldsymbol{\epsilon}, \tag{27}$$

where $\mathbf{x} \sim q(\mathbf{x})$ and $\boldsymbol{\epsilon} \sim p(\boldsymbol{\epsilon})$. The marginal distribution at time $t$ is

$$q_t(\mathbf{x}_t) = \iint \delta(\mathbf{x}_t - I(\mathbf{x}, \boldsymbol{\epsilon}, t))q(\mathbf{x})p(\boldsymbol{\epsilon})\,\mathrm{d}\mathbf{x}\,\mathrm{d}\boldsymbol{\epsilon}. \tag{28}$$

Under the DDIM formulation, the trajectory is deterministic conditioned on $\mathbf{x}$. Therefore, for arbitrary $s, t \in [0, 1]$, the state $\mathbf{x}_s$ is uniquely determined by $\mathbf{x}$ and $\mathbf{x}_t$, and the conditional distribution can be written as

$$q_{s|t}(\mathbf{x}_s|\mathbf{x}, \mathbf{x}_t) = \delta\left(\mathbf{x}_s - \left[a_s\mathbf{x} + b_s\frac{\mathbf{x}_t - a_t\mathbf{x}}{b_t}\right]\right). \tag{29}$$

Substituting the DDIM conditional distribution into the right-hand side of the marginal-preserving equality yields

$$\iint q_{s|t}(\mathbf{x}_s|\mathbf{x}, \mathbf{x}_t)q_t(\mathbf{x}, \mathbf{x}_t)\,\mathrm{d}\mathbf{x}_t\,\mathrm{d}\mathbf{x}$$
$$= \iiint \delta\left(\mathbf{x}_s - \left[a_s\mathbf{x} + \frac{b_s}{b_t}(\mathbf{x}_t - a_t\mathbf{x})\right]\right)\delta(\mathbf{x}_t - (a_t\mathbf{x} + b_t\boldsymbol{\epsilon}))q(\mathbf{x})p(\boldsymbol{\epsilon})\,\mathrm{d}\boldsymbol{\epsilon}\,\mathrm{d}\mathbf{x}_t\,\mathrm{d}\mathbf{x}. \tag{30}$$

Using the DDIM construction,

$$a_s\mathbf{x} + \frac{b_s}{b_t}\left((a_t\mathbf{x} + b_t\boldsymbol{\epsilon}) - a_t\mathbf{x}\right) = a_s\mathbf{x} + b_s\boldsymbol{\epsilon} = I(\mathbf{x}, \boldsymbol{\epsilon}, s), \tag{31}$$

we obtain

$$\iint q_{s|t}(\mathbf{x}_s|\mathbf{x}, \mathbf{x}_t)q_t(\mathbf{x}, \mathbf{x}_t)\,\mathrm{d}\mathbf{x}_t\,\mathrm{d}\mathbf{x}$$
$$= \iint \delta(\mathbf{x}_s - I(\mathbf{x}, \boldsymbol{\epsilon}, s))q(\mathbf{x})p(\boldsymbol{\epsilon})\,\mathrm{d}\boldsymbol{\epsilon}\,\mathrm{d}\mathbf{x} = q_s(\mathbf{x}_s). \tag{32}$$

Therefore, DDIM interpolants intrinsically satisfy the marginal-preserving equality.

Since DDIM interpolants satisfy the marginal-preserving equality, we can reuse the sampled state $\mathbf{x}_t$ to construct $\mathbf{x}_r$ through the conditional distribution

$$\mathbf{x}_r \sim q_{r|t}(\mathbf{x}_r \mid \mathbf{x}, \mathbf{x}_t),$$

instead of independently sampling $\mathbf{x}_r$ from the forward flow

$$\mathbf{x}_r = \alpha_r\mathbf{x} + \sigma_r\boldsymbol{\epsilon}.$$

In particular, we use the deterministic DDIM mapping

$$\mathbf{x}_r = \text{DDIM}(\mathbf{x}_t, \mathbf{x}, r, t),$$

which reduces the variance of empirical estimation.

More generally, for any marginal-preserving interpolant, the marginal distribution of $\mathbf{x}_r$ remains unchanged under conditional resampling:

$$
\begin{aligned}
q_r(\mathbf{x}_r) = q_{r|t}(\mathbf{x}_r) &= \iint q_{r|t}(\mathbf{x}_r \mid \mathbf{x}, \mathbf{x}_t) q_t(\mathbf{x} \mid \mathbf{x}_t) q_t(\mathbf{x}_t) \, \mathrm{d}\mathbf{x} \, \mathrm{d}\mathbf{x}_t \\
&= \iiint q_{r|t}(\mathbf{x}_r \mid \mathbf{x}, \mathbf{x}_t) q_t(\mathbf{x}_t \mid \mathbf{x}, \boldsymbol{\epsilon}) q(\mathbf{x}) p(\boldsymbol{\epsilon}) \, \mathrm{d}\boldsymbol{\epsilon} \, \mathrm{d}\mathbf{x} \, \mathrm{d}\mathbf{x}_t.
\end{aligned}
$$

Therefore, sampling $(\mathbf{x}, \mathbf{x}_t)$ first and then drawing

$$
\mathbf{x}_r \sim q_{r|t}(\mathbf{x}_r \mid \mathbf{x}, \mathbf{x}_t)
$$

still preserves the target marginal distribution $q_r(\mathbf{x}_r)$.

Next, we try to simplify the objective function 10. Given MMD loss function we defined in 3, we have:

$$
\begin{aligned}
\mathcal{L}_D(\theta) &= \mathbb{E}_{s,t}\left[\left\|\mathbb{E}_{\mathbf{x}_t}[k(f_{s,t}^\theta(\mathbf{x}_t), \cdot)] - \mathbb{E}_{\mathbf{x}_r}[k(f_{s,r}^{\theta^-}(\mathbf{x}_r), \cdot)]\right\|_{\mathcal{H}}^2\right] \\
&= \mathbb{E}_{s,t}\left[\left\|\mathbb{E}_{\mathbf{x}_t,\mathbf{x}_r}[k(f_{s,t}^\theta(\mathbf{x}_t), \cdot) - k(f_{s,r}^{\theta^-}(\mathbf{x}_r), \cdot)]\right\|_{\mathcal{H}}^2\right] \quad \text{(reuse the sample } \mathbf{x}_t) \\
&= \mathbb{E}_{s,t}\left[\left\langle\mathbb{E}_{\mathbf{x}_t,\mathbf{x}_r}[k(f_{s,t}^\theta(\mathbf{x}_t), \cdot) - k(f_{s,r}^{\theta^-}(\mathbf{x}_r), \cdot)], \mathbb{E}_{\mathbf{x}_t',\mathbf{x}_r'}[k(f_{s,t}^\theta(\mathbf{x}_t'), \cdot) - k(f_{s,r}^{\theta^-}(\mathbf{x}_r'), \cdot)]\right\rangle_{\mathcal{H}}\right] \\
&= \mathbb{E}_{s,t}\Bigg[\mathbb{E}_{\mathbf{x}_t,\mathbf{x}_r,\mathbf{x}_t',\mathbf{x}_r'}\Big[\left\langle k(f_{s,t}^\theta(\mathbf{x}_t), \cdot), k(f_{s,t}^\theta(\mathbf{x}_t'), \cdot)\right\rangle_{\mathcal{H}} + \left\langle k(f_{s,r}^{\theta^-}(\mathbf{x}_r), \cdot), k(f_{s,r}^{\theta^-}(\mathbf{x}_r'), \cdot)\right\rangle_{\mathcal{H}} \\
&\qquad - \left\langle k(f_{s,t}^\theta(\mathbf{x}_t), \cdot), k(f_{s,r}^{\theta^-}(\mathbf{x}_r'), \cdot)\right\rangle_{\mathcal{H}} - \left\langle k(f_{s,t}^\theta(\mathbf{x}_t'), \cdot), k(f_{s,r}^{\theta^-}(\mathbf{x}_r), \cdot)\right\rangle_{\mathcal{H}}\Big]\Bigg] \\
&= \mathbb{E}_{\mathbf{x}_t,\mathbf{x}_r,\mathbf{x}_t',\mathbf{x}_r',s,t}\Bigg[\Big[k(f_{s,t}^\theta(\mathbf{x}_t), f_{s,t}^\theta(\mathbf{x}_t')) + k(f_{s,r}^{\theta^-}(\mathbf{x}_r), f_{s,r}^{\theta^-}(\mathbf{x}_r')) \\
&\qquad - k(f_{s,t}^\theta(\mathbf{x}_t), f_{s,r}^{\theta^-}(\mathbf{x}_r')) - k(f_{s,t}^\theta(\mathbf{x}_t'), f_{s,r}^{\theta^-}(\mathbf{x}_r))\Big]\Bigg] \quad \text{(reproducing property)}
\end{aligned}
$$

If we select kernel function $k(x, y) = -\|x - y\|_2^2$, $\mathbf{x}_t = \mathbf{x}_t'$, $\mathbf{x}rt = \mathbf{x}_r'$, and $s = \epsilon$, we can see that the MMD loss function is exactly the same as the consistency training loss in the form of $\mathcal{L}_{CT} = \mathbb{E}_{\mathbf{x}_t,\mathbf{x},t}\left[\|f_\theta(\mathbf{x}_t, t) - f_{\theta^-}(\mathbf{x}_r, r)\|^2\right]$. From a moment-matching perspective, $\mathcal{L}_{CT}$ significantly deviates from a proper divergence between distributions, as (i) $\mathcal{L}_{CT}$ only conduct multi-particle estimation, which may pause mode collapse and training instability and (ii) this kernel is not positive definite kernel and can only match the first order (Expectation) and second order (Variance) moment, which can hardly match two high-dimensional distributions.

As proposed in (Gretton et al., 2012), MMD loss can be estimated with V-statistics by instantiating a matrix of size $M \times M$ such that a batch of $B$ samples, $\{x^{(i)}\}_{i=1}^B$, is separated into groups of $M$ (assume $B$ is divisible by $M$) particles $\{x^{(i,j)}\}_{i=1,j=1}^{B/M,M}$ where each group share a $(s^i, r^i, t^i)$ sample. We also add the weight function $w(s, t) = \frac{1}{\alpha_t^2 + \sigma_t^2}$ to improve the performance. The estimation of the MMD loss can be represented as:

$$
\begin{aligned}
\hat{\mathcal{L}}_D(\theta) = \frac{1}{B/M}\sum_{i=1}^{B/M} w(s^i, t^i)\frac{1}{M^2}\sum_{j=1}^{M}\sum_{k=1}^{M}\Big[&k(f_{s^i,t^i}^\theta(\mathbf{x}_{t^i}^{(i,j)}), f_{s^i,t^i}^\theta(\mathbf{x}_{t^i}^{(i,k)})) + k(f_{s^i,r^i}^{\theta^-}(\mathbf{x}_{r^i}^{(i,j)}), f_{s^i,r^i}^{\theta^-}(\mathbf{x}_{r^i}^{(i,k)})) \\
&- 2k(f_{s^i,t^i}^\theta(\mathbf{x}_{t^i}^{(i,j)}), f_{s^i,r^i}^{\theta^-}(\mathbf{x}_{r^i}^{(i,k)}))\Big]
\end{aligned} \tag{33}
$$

This is the final loss firm that we utilize to train the model. Compared to the loss function in consistency models, this empirical MMD loss can capture the high-order moment information of target distribution via a kernel function, and thus can match the distribution more accurately.

## B. Full Algorithm

A more detailed training algorithm is presented in Algorithm 3. We utilized RBF kernel as default kernel function $k(\cdot, \cdot)$, other kernels such as Laplacian kernel are also applicable.

---

**Algorithm 3** MoMa QL Training

---

**Input:** offline dataset $\mathcal{D}$, model $f^\theta$, critic networks $Q_{\phi_1}, Q_{\phi_2}$, data distribution $q(\mathbf{x})$, prior distribution $\mathcal{N}(0, \sigma_d^2 I)$, time distribution $p(t)$ and $p(s|t)$, DDIM interpolator $\text{DDIM}(\mathbf{x}_t, \mathbf{x}, s, t)$ and its flow coefficients $\alpha_t, \sigma_t$, mapping function $r(s,t)$, kernel function $k(\cdot, \cdot)$, weighting function $w(s,t)$, batch size $B$, particle number $M$

Initialize policy network $\pi_\theta$, critic networks $Q_{\phi_1}, Q_{\phi_2}$

Initialize target network parameters: $\theta^{\mathsf{T}} \leftarrow \theta, \phi_1^{\mathsf{T}} \leftarrow \phi_1, \phi_2^{\mathsf{T}} \leftarrow \phi_2$

**while** model not converged **do**

    Sample a batch of batch $\mathcal{B} = \{(\mathbf{s}, \mathbf{a}, r, \mathbf{s}')\} \subseteq \mathcal{D}$

    // Q-value Update

    Update $Q_{\phi_1}, Q_{\phi_2}$ via Eq. (11);

    // Policy Update

    Split $\mathcal{B}$ into $B/M$ groups, sample prior $\varepsilon$ to get the splited dataset as $\{(\mathbf{a}^{(i,j)}, \mathbf{s}^{(i,j)}, \varepsilon^{(i,j)})\}_{i=1,j=1}^{B/M,M}$

    For each group, sample $\{(s^i, t^i)\}_{i=1}^{B/M}$ and $r^i = r(s^i, t^i)$ for each $i$. This results in a tuple $\{(s^i, r^i, t^i)\}_{i=1}^{B/M}$

    $\mathbf{a}_{t^i}^{(i,j)} \leftarrow \text{DDIM}(\varepsilon^{(i,j)}, \mathbf{a}^{(i,j)}, t^i, 1) = \alpha_{t^i} \mathbf{a}^{(i,j)} + \sigma_{t^i} \varepsilon^{(i,j)}, \forall (i,j)$

    $\mathbf{a}_{r^i}^{(i,j)} \leftarrow \text{DDIM}(\mathbf{a}_{t^i}^{(i,j)}, \mathbf{a}^{(i,j)}, r^i, t^i), \forall (i,j)$

    Calculate $\hat{\mathcal{L}}_D(\theta)$ using model $f^\theta$ in Eq. 33 with inputting $\mathbf{s}^{(i,j)}$ into network as condition

    Sample an action $\hat{\mathbf{a}}^{(i,j)}$ from $f^\theta$ given $\mathbf{s}^{(i,j)}$

    $\theta \leftarrow$ optimizer step by minimizing $\hat{\mathcal{L}}_\pi(\theta) := \hat{\mathcal{L}}_D(\theta_n) - \eta * \min\left(Q_{\phi_1}(\mathbf{s}^{(i,j)}, \hat{\mathbf{a}}^{(i,j)}), Q_{\phi_2}(\mathbf{s}^{(i,j)}, \hat{\mathbf{a}}^{(i,j)})\right)$

    // Target Update

    Update target: $\theta^{\mathsf{T}} \leftarrow \alpha \theta^{\mathsf{T}} + (1-\alpha)\theta, \phi_i^{\mathsf{T}} \leftarrow \alpha \phi_i^{\mathsf{T}} + (1-\alpha)\phi_i, i \in \{1,2\}$;

**end while**

**return** $f_\theta, Q_{\phi_1}, Q_{\phi_2}$

---

## C. Experiments

### C.1. Experimental Hyperparameters

We provide the detailed hyperparameters used for training MoMa QL in Table 4. We use a consistent set of hyperparameters across most D4RL Gym locomotion tasks to demonstrate the robustness of our method. For the Kitchen and Adroit environments, minor adjustments (e.g., learning rate or epoch count) were made to accommodate the different task complexities, as noted.

### C.2. Extended Offline RL Results and Analysis

This appendix provides a detailed breakdown of MoMa QL's performance across all individual D4RL datasets. Table 5 presents the full comparison with standard deviations included.

#### C.2.1. DETAILED ANALYSIS

**Gym Locomotion Tasks.** MoMa QL demonstrates remarkable efficacy on suboptimal and diverse datasets. On `halfcheetah-m`, our method achieves a score of $\mathbf{72.6 \pm 1.1}$, surpassing the best baseline (Consistency-AC, 69.1) by approximately $5\%$. Similarly, on the challenging `walker2d-mr` dataset, MoMa QL attains $\mathbf{104.3 \pm 3.3}$, outperforming Diffusion-QL (95.5) by a factor of 1.09. These results underscore the robustness of our moment-matching objective in extracting optimal policies from noisy data. While Diffusion-QL holds a slight edge on `hopper-me` (111.1 vs. our $88.9 \pm 9.6$), MoMa QL consistently dominates on medium and medium-replay datasets.

**Adroit Manipulation Tasks.** The detailed results reveal specific strengths of our method. On `door-expert-v1`, MoMa QL achieves a near-perfect score of $\mathbf{108 \pm 1}$, narrowly edging out ReBRAC (106) and IQL (107). In the `pen-cloned-v1` task, our method reaches $\mathbf{104 \pm 8}$, matching the performance of ReBRAC. However, on human datasets like `hammer-human`, all methods struggle (scores $< 5$), indicating a general difficulty in learning from these demonstrations, though MoMa QL remains competitive with a score of $\mathbf{4 \pm 0}$.

**Kitchen Tasks.** Our method shines particularly on the `kitchen-complete` task, achieving a score of $\mathbf{91.3 \pm 3.6}$, which is an improvement of roughly $1.09\times$ over Diffusion-QL (84.0) and $1.11\times$ over $\mathcal{X}$-QL (82.4). This suggests that for tasks

*Table 4.* Hyperparameters for MoMa QL on D4RL benchmarks. Unless otherwise specified, these values are used for all Gym locomotion tasks.

| Hyperparameter | Value |
|---|---|
| *Optimization* | |
| Optimizer | Adam |
| Learning Rate | $1 \times 10^{-3}$ (Gym), $5 \times 10^{-4}$ (Kitchen) |
| Batch Size | 256 |
| Gradient Norm Clip | 8.0 |
| Training Epochs | 500 (Gym), 1000 (Kitchen) |
| Steps per Epoch | 1000 |
| $\eta$ (Q-learning Weight) | 0.5 |
| Target Network Update Rate ($\tau$) | 0.005 |
| Discount Factor ($\gamma$) | 0.99 |
| *MMD Noise Schedule* | |
| MMD Kernel $a$ | 4 |
| MMD Kernel $b$ | 2 |
| MMD Kernel $k$ | 8 |
| MMD Sigma ($\sigma_{\text{MMD}}$) | 1.2 |
| Noise Schedule | Flow Matching (FM) |
| $p_{\text{mean}}$ | -0.8 |
| $p_{\text{std}}$ | 1.5 |
| $\sigma_{\text{data}}$ | 0.5 |
| ODE Solver | Euler |
| *Architecture* | |
| Model Type | MLP |
| Hidden Dim | 256 |
| Number of Layers | 3 |

requiring the completion of full sequential goals, MoMa QL's generative policy is highly effective. On partial and mixed datasets, performance is comparable to leading baselines, maintaining stable results around the 60.0 mark.

## C.3. Extended Offline-to-Online RL Results

This appendix provides comprehensive analysis of MoMa QL's performance in the offline-to-online setting, including detailed per-dataset results.

Table 6 presents the complete offline-to-online results across all datasets with standard deviations. For each task, we compare against strong baselines including Diffusion-QL and Consistency-AC.

**Analysis.** MoMa QL achieves an average score of **101.4** across all tasks, demonstrating superior final performance.

- On **Medium-Replay** datasets, our method excels, achieving **113.5** on Hopper-mr and **117.9** on Walker2d-mr, significantly outperforming Diffusion-QL (68.4 and 95.7, respectively).

- On **Medium-Expert** datasets, MoMa QL remains competitive, with particularly strong results on Walker2d-me (**118.0**).

- The transition from offline to online is stable, with no catastrophic forgetting observed, a common issue in fine-tuning generative policies.

## C.4. Detailed Ablation Studies

In this section, we provide a comprehensive analysis of the hyperparameters used in MoMa QL. We systematically vary one parameter at a time while keeping others fixed.

*Table 5.* Detailed offline RL results on D4RL benchmarks by dataset. We report normalized scores averaged over 5 random seeds with standard deviations where available. Bold values indicate the best performance in each row.

| Gym Tasks | CQL | IQL | $\mathcal{X}$-QL | ARQ | IDQL-A | Diffusion-QL | Consistency-AC | MoMa QL (Ours) |
|---|---|---|---|---|---|---|---|---|
| halfcheetah-m | 44.0 | 47.4 | 48.3 | 45±0.3 | 51.0 | 51.1±0.5 | 69.1±0.7 | **72.6±1.1** |
| hopper-m | 58.5 | 66.3 | 74.2 | 61±0.4 | 65.4 | 90.5±4.6 | 80.7±10.5 | **104.2±2.3** |
| walker2d-m | 72.5 | 78.3 | 84.2 | 81±0.7 | 82.5 | 87.0±0.9 | 83.1±0.3 | **95.6±0.4** |
| halfcheetah-mr | 45.5 | 44.2 | 45.2 | 42±0.3 | 45.9 | 47.8±0.3 | 58.7±3.9 | **63.3±0.7** |
| hopper-mr | 95.0 | 94.7 | 100.7 | 81±24.2 | 92.1 | 101.3±0.6 | 99.7±0.5 | **106.5±1.4** |
| walker2d-mr | 77.2 | 73.9 | 82.2 | 66±7.0 | 85.1 | 95.5±1.5 | 79.5±3.6 | **104.3±3.3** |
| halfcheetah-me | 91.6 | 86.7 | 94.2 | 91±0.7 | 95.9 | 96.8±0.3 | 84.3±4.1 | **106.9±0.4** |
| hopper-me | 105.4 | 91.5 | 111.2 | 110±0.9 | 108.6 | **111.1±1.3** | 100.4±3.5 | 88.9±9.6 |
| walker2d-me | 108.8 | 109.6 | 112.7 | 109±0.5 | 112.7 | 110.1±0.3 | 110.4±0.7 | **117.5±0.8** |
| Average | 77.6 | 77.0 | 83.7 | 76.2 | 82.1 | 87.9 | 85.1 | **95.5** |

| Adroit Tasks | BC | IQL | ReBRAC | IDQL | SRPO | CAC | FAWAC | FBRAC | IFQL | FQL | MoMa QL (Ours) |
|---|---|---|---|---|---|---|---|---|---|---|---|
| pen-human-v1 | 71 | 78 | **103** | 76±10 | 69±7 | 64±8 | 67±5 | 77±7 | 71±12 | 53±6 | 88±7 |
| pen-cloned-v1 | 52 | 83 | 103 | 64±7 | 61±7 | 56±10 | 62±10 | 67±9 | 80±11 | 74±11 | **104±8** |
| pen-expert-v1 | 110 | 128 | **152** | 140±6 | 134±4 | 103±9 | 118±6 | 119±7 | 139±5 | 142±6 | 134±5 |
| door-human-v1 | 2 | 3 | -0 | 6±2 | 3±3 | 5±2 | 2±1 | 4±2 | 7±2 | 0±0 | **7±1** |
| door-cloned-v1 | -0 | **3** | 0 | 0±0 | 0±0 | 1±0 | 0±1 | 0±0 | 2±2 | 2±1 | 0±0 |
| door-expert-v1 | 105 | 107 | 106 | 105±1 | 105±0 | 98±3 | 103±1 | 104±1 | 104±2 | 104±1 | **108±1** |
| hammer-human-v1 | 3 | 2 | 0 | 2±1 | 1±1 | 2±0 | 2±1 | 2±1 | 3±1 | 1±1 | **4±0** |
| hammer-cloned-v1 | 1 | 2 | 5 | 2±1 | 2±1 | 1±1 | 1±0 | 2±1 | 2±1 | **11±9** | 3±1 |
| hammer-expert-v1 | 127 | 129 | **134** | 125±4 | 127±0 | 92±11 | 118±3 | 119±9 | 117±9 | 125±3 | 126±2 |
| relocate-human-v1 | 0 | 0 | 0 | 0±0 | 0±0 | 0±0 | 0±0 | 0±0 | 0±0 | 0±0 | 0±0 |
| relocate-cloned-v1 | -0 | 0 | **2** | -0±0 | -0±0 | -0±0 | -0±0 | 1±1 | -0±0 | -0±0 | 0±0 |
| relocate-expert-v1 | **108** | 106 | **108** | 107±1 | 106±2 | 93±6 | 105±3 | 105±2 | 104±3 | 107±1 | **108±1** |
| Average | 48.3 | 53.4 | 55.4 | 52.3 | 50.7 | 42.9 | 48.2 | 50.0 | 52.4 | 51.6 | **56.7** |

| Kitchen Tasks | CQL | IQL | $\mathcal{X}$-QL | ARQ | Diffusion-QL | Consistency-AC | MoMa QL (Ours) |
|---|---|---|---|---|---|---|---|
| kitchen-complete | 43.8 | 62.5 | 82.4 | 37±14.2 | 84.0±7.4 | 51.9±6.0 | **91.3±3.6** |
| kitchen-partial | 49.8 | 46.3 | **73.7** | 50±5.0 | 60.5±6.9 | 38.2±1.8 | 63.3±7.6 |
| kitchen-mixed | 51.0 | 51.0 | **62.5** | 39±9.4 | **62.6±5.1** | 45.8±1.5 | 58.8±6.3 |
| Average | 48.2 | 53.3 | 72.9 | 42.0 | 69.0 | 45.3 | **73.1** |

### C.4.1. HYPERPARAMETER DEFINITIONS

- $a$: Controls the power of $\alpha_t$ in the kernel weighting function $\alpha_t^a/(\alpha_t^2 + \sigma_t^2)$. A larger $a$ emphasizes cleaner samples (small noise), allowing the model to capture fine details.

- $b$: Appears in the sigmoid weighting term $(b - \log \text{SNR}_t)$. It shifts the center of the weighting function. A higher $b$ focuses on noisier samples, while a lower $b$ focuses on cleaner samples.

- $\eta$: The regularizing weight for the Q-value loss. It controls how much we trust the Q-function versus the behavioral distribution.

- **Number of Steps**: The number of denoising steps used during action sampling.

- $P_{\text{mean}}$: The mean of the log-normal distribution for sampling time steps $t$ during training. a negative value biases sampling towards $t = 0$ (clean data).

- $P_{\text{std}}$: The standard deviation of the time step distribution. A larger value covers a wider range of noise levels.

*Table 6.* Detailed offline-to-online RL results on D4RL benchmarks. For each method, we report the final online score. All results are averaged over 5 random seeds with standard deviations. Bold values indicate the best performance.

| Gym Tasks | SAC | AWAC | ACA | Diffusion-QL | Consistency-AC | MoMa QL (Ours) |
|---|---|---|---|---|---|---|
| halfcheetah-m | 75.2 | 50.5 | 66.6 | **99.6±2.3** | 98.7±1.8 | 83.1±1.3 |
| hopper-m | 73.4 | 97.5 | 96.5 | 77.2±25.6 | 60.5±8.6 | **104.3±4.6** |
| walker2d-m | 79.6 | 1.9 | 74.7 | **118.3±5.8** | 108.9±3.0 | 99.1±2.3 |
| halfcheetah-mr | 68.9 | 46.8 | 59.0 | **96.3±3.9** | 80.7±10.5 | 80.9±1.2 |
| hopper-mr | 74.0 | 96.0 | 85.5 | 68.4±20.3 | 74.6±25.1 | **113.5±3.6** |
| walker2d-mr | 85.4 | 80.8 | 85.2 | 95.7±18.8 | 102.0±11.6 | **117.9±4.4** |
| halfcheetah-me | 82.2 | 68.8 | 93.7 | 103.9±2.2 | 99.6±4.1 | **107.1±6.1** |
| hopper-me | 65.4 | 73.1 | **98.0** | 71.7±31.1 | 65.4±5.7 | 89.1±8.8 |
| walker2d-me | 87.2 | 45.2 | 110.5 | 117.0±6.3 | 101.8±13.3 | **118.0±7.3** |
| Average | 76.8 | 62.3 | 85.5 | 94.2 | 88.0 | **101.4** |

### C.4.2. ADDITIONAL RESULTS

We present the full ablation results for all parameters across HalfCheetah, Hopper, and Walker2d environments.

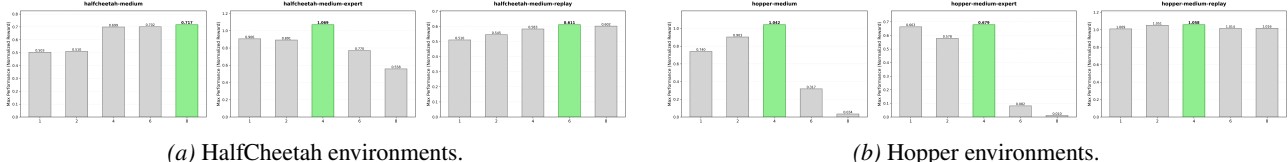

*(a)* HalfCheetah environments.  *(b)* Hopper environments.

*Figure 6.* Ablation study on parameter $a$.

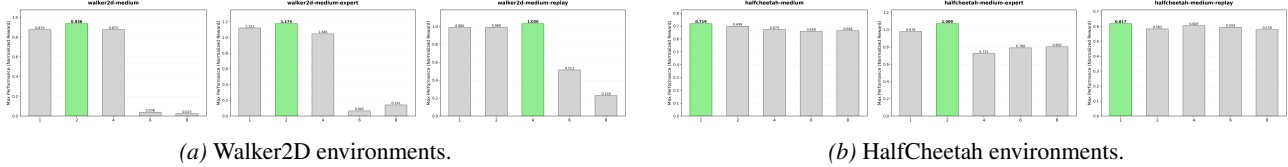

*(a)* Walker2D environments.  *(b)* HalfCheetah environments.

*Figure 7.* Ablation study on parameters $a$ (Walker2D) and $b$ (HalfCheetah).

## C.5. Detailed Computational Cost Results

### C.5.1. BASELINE COMPARISON

Table 7 reports the detailed computational cost for MoMa QL compared with Diffusion-BC and Consistency-BC. We report both **Training Time** (seconds per 1,000 steps) and **Inference Time** (seconds per episode) for MoMa QL with 2 sampling steps. Baseline training times are converted from reported values (seconds per 500-step epoch $\times 2$) for fair comparison.

MoMa QL maintains a consistent training cost ($\approx 21 - 23$s per 1k steps) across all tasks, demonstrating superior scalability compared to baselines which slow down significantly on higher-dimensional tasks (Adroit, Kitchen). All methods use comparable batch sizes and hardware configurations.

### C.5.2. IMPACT OF SAMPLING STEPS

Tables 8, 9, and 10 present the detailed training time (seconds per 1,000 steps) as a function of the number of sampling steps for MoMa QL across all D4RL tasks. The results demonstrate how computational cost scales with sampling steps while maintaining efficiency.

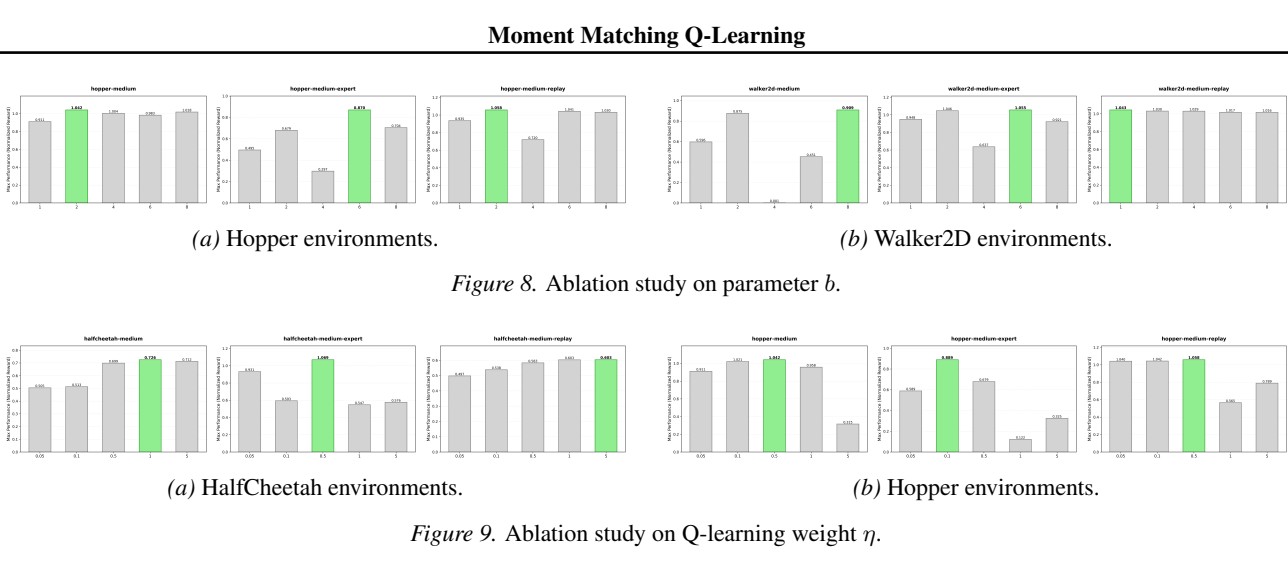

*(a)* Hopper environments.  *(b)* Walker2D environments.

*Figure 8.* Ablation study on parameter *b*.

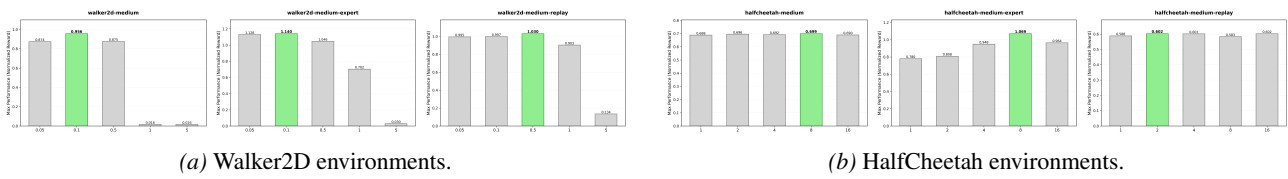

*(a)* HalfCheetah environments.  *(b)* Hopper environments.

*Figure 9.* Ablation study on Q-learning weight $\eta$.

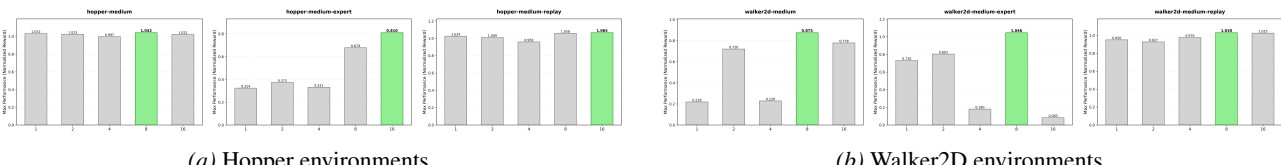

*(a)* Walker2D environments.  *(b)* HalfCheetah environments.

*Figure 10.* Ablation study on $\eta$ (Walker2D) and steps (HalfCheetah).

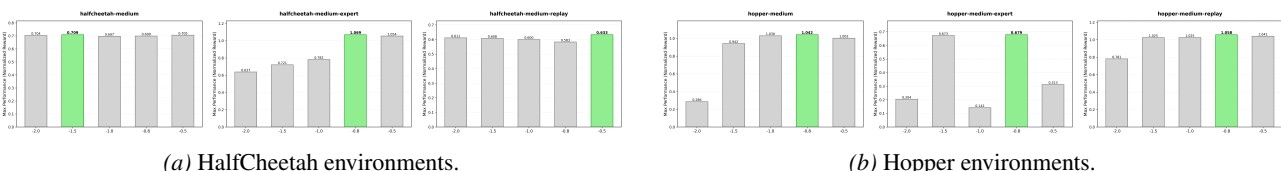

*(a)* Hopper environments.  *(b)* Walker2D environments.

*Figure 11.* Ablation study on num steps.

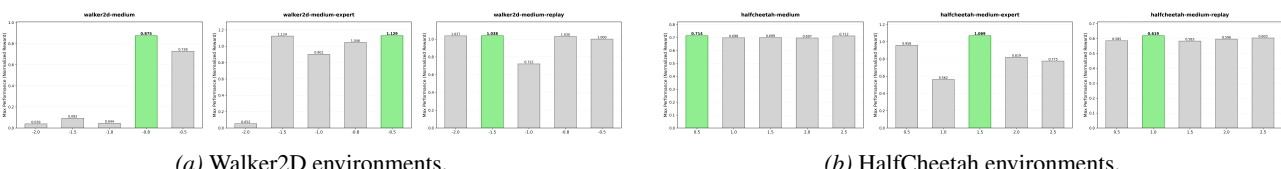

*(a)* HalfCheetah environments.  *(b)* Hopper environments.

*Figure 12.* Ablation study on $P_{\mathrm{mean}}$.

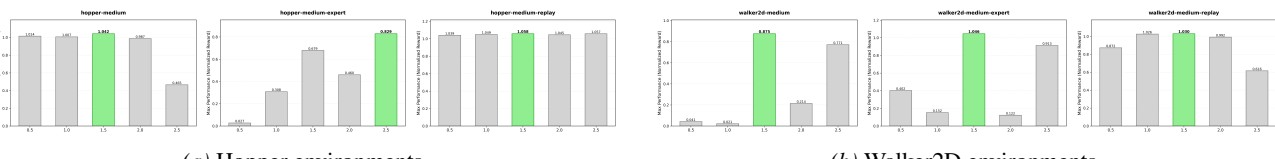

*(a)* Walker2D environments.  *(b)* HalfCheetah environments.

*Figure 13.* Ablation study on $P_{\mathrm{mean}}$ (Walker2D) and $P_{\mathrm{std}}$ (HalfCheetah).

*(a)* Hopper environments.  *(b)* Walker2D environments.

*Figure 14.* Ablation study on $P_{\mathrm{std}}$.

*Table 7.* Computational cost on D4RL benchmarks. We compare Training Time (s/1k steps) against baselines including Diffusion-BC, Consistency-BC, Diffusion-QL, and Consistency-AC. For fair comparison, baselines are normalized to s/1k steps: Diffusion-QL and Consistency-AC are converted from hours/1M steps, while Diffusion-BC and Consistency-BC are converted from s/500-step epoch. Confidence intervals are omitted.

| Task | Training Time (s / 1k steps) | | | | | Inference (Ours) |
|---|---|---|---|---|---|---|
| | Diffusion-BC | Consistency-BC | Diffusion-QL | Consistency-AC | **MoMa QL (Ours)** | s / episode |
| *Gym Locomotion* | | | | | | |
| halfcheetah-m | 135.9 | 86.1 | 41.6 | 34.6 | **21.78** | 2.89 |
| hopper-m | 122.0 | 76.0 | 32.3 | 24.8 | **21.33** | 0.56 |
| walker2d-m | 136.3 | 87.2 | 33.0 | 29.5 | **21.33** | 0.42 |
| halfcheetah-mr | 134.1 | 85.5 | 33.0 | 27.8 | **21.27** | 2.79 |
| hopper-mr | 129.8 | 76.1 | 29.6 | 26.1 | **22.23** | 0.53 |
| walker2d-mr | 132.2 | 85.4 | 31.9 | 26.4 | **21.01** | 0.44 |
| halfcheetah-me | 135.5 | 87.2 | 33.3 | 30.5 | **20.63** | 2.93 |
| hopper-me | 126.1 | 77.1 | 29.8 | 26.9 | **21.13** | 0.19 |
| walker2d-me | 138.2 | 87.8 | 35.8 | 33.9 | **22.84** | 0.80 |
| *Gym Average* | 132.2 | 83.1 | 33.4 | 28.9 | **21.51** | 1.28 |
| *Adroit Manipulation* | | | | | | |
| pen-human-v1 | 184.4 | 93.9 | - | - | **21.05** | 0.38 |
| pen-cloned-v1 | 189.3 | 100.7 | - | - | **22.06** | 0.42 |
| pen-expert-v1 | - | - | - | - | **21.25** | 0.39 |
| door-human-v1 | - | - | - | - | **21.90** | 0.69 |
| door-cloned-v1 | - | - | - | - | **21.72** | 0.71 |
| door-expert-v1 | - | - | - | - | **22.47** | 0.73 |
| hammer-human-v1 | - | - | - | - | **21.26** | 0.76 |
| hammer-cloned-v1 | - | - | - | - | **21.23** | 0.76 |
| hammer-expert-v1 | - | - | - | - | **21.33** | 0.77 |
| relocate-human-v1 | - | - | - | - | **21.51** | 0.71 |
| relocate-cloned-v1 | - | - | - | - | **21.39** | 0.71 |
| relocate-expert-v1 | - | - | - | - | **22.76** | 0.76 |
| *Adroit Average* | 186.8 | 97.3 | - | - | **21.66** | 0.67 |
| *Fanka Kitchen* | | | | | | |
| kitchen-complete | 193.5 | 133.4 | - | - | **23.11** | 1.94 |
| kitchen-partial | 188.5 | 123.7 | - | - | **21.23** | 1.95 |
| kitchen-mixed | 186.0 | 133.2 | - | - | **22.64** | 2.08 |
| *Kitchen Average* | 189.4 | 130.1 | - | - | **22.33** | 1.99 |

Key observations: The training time exhibits nearly linear scaling with respect to the number of sampling steps across all domains. For single-step sampling ($n = 1$), the average training time is 17.70s (Gym), 17.64s (Adroit), and 17.71s (Kitchen). This increases to 72.56s, 73.12s, and 70.94s respectively for 16-step sampling, representing approximately a $4\times$ increase for a $16\times$ increase in steps. This sub-linear scaling demonstrates the computational efficiency of our MMD-based approach. Notably, task complexity has minimal impact on training time—high-dimensional Kitchen and Adroit tasks require similar computational resources as lower-dimensional Gym tasks, highlighting MoMa QL's superior scalability compared to diffusion-based baselines.

*Table 8.* Training time (seconds per 1,000 steps) for different numbers of sampling steps on Gym locomotion tasks.

| Task | 1 step | 2 steps | 4 steps | 8 steps | 16 steps |
|------|--------|---------|---------|---------|----------|
| halfcheetah-m | 17.10 | 21.78 | 28.18 | 43.64 | 70.37 |
| hopper-m | 18.50 | 21.33 | 27.53 | 44.40 | 70.57 |
| walker2d-m | 17.19 | 21.33 | 30.04 | 44.13 | 74.62 |
| halfcheetah-mr | 17.81 | 21.27 | 30.80 | 43.51 | 70.74 |
| hopper-mr | 18.01 | 22.23 | 28.75 | 44.73 | 72.10 |
| walker2d-mr | 17.94 | 21.01 | 29.70 | 42.21 | 69.42 |
| halfcheetah-me | 17.04 | 20.63 | 28.79 | 43.66 | 77.23 |
| hopper-me | 17.74 | 21.13 | 28.86 | 42.61 | 74.38 |
| walker2d-me | 17.95 | 22.84 | 29.96 | 43.41 | 73.63 |
| *Average* | 17.70 | 21.51 | 29.18 | 43.59 | 72.56 |

*Table 9.* Training time (seconds per 1,000 steps) for different numbers of sampling steps on Adroit manipulation tasks.

| Task | 1 step | 2 steps | 4 steps | 8 steps | 16 steps |
|------|--------|---------|---------|---------|----------|
| pen-human-v1 | 17.40 | 21.05 | 28.70 | 44.41 | 73.48 |
| pen-cloned-v1 | 17.36 | 22.06 | 28.07 | 44.59 | 73.02 |
| pen-expert-v1 | 17.40 | 21.25 | 29.09 | 44.38 | 71.33 |
| door-human-v1 | 17.34 | 21.90 | 28.87 | 44.03 | 75.82 |
| door-cloned-v1 | 17.56 | 21.72 | 28.49 | 41.80 | 72.85 |
| door-expert-v1 | 17.33 | 22.47 | 29.68 | 42.75 | 74.08 |
| hammer-human-v1 | 18.41 | 21.26 | 28.48 | 43.61 | 70.04 |
| hammer-cloned-v1 | 18.34 | 21.23 | 29.06 | 44.51 | 70.87 |
| hammer-expert-v1 | 17.97 | 21.33 | 29.70 | 43.71 | 70.66 |
| relocate-human-v1 | 17.71 | 21.51 | 28.12 | 45.26 | 78.24 |
| relocate-cloned-v1 | 17.39 | 21.39 | 28.32 | 43.34 | 72.31 |
| relocate-expert-v1 | 17.45 | 22.76 | 29.73 | 44.01 | 74.79 |
| *Average* | 17.64 | 21.66 | 28.86 | 43.87 | 73.12 |

*Table 10.* Training time (seconds per 1,000 steps) for different numbers of sampling steps on Franka Kitchen tasks.

| Task | 1 step | 2 steps | 4 steps | 8 steps | 16 steps |
|------|--------|---------|---------|---------|----------|
| kitchen-complete | 18.88 | 23.11 | 28.40 | 44.60 | 71.37 |
| kitchen-partial | 16.98 | 21.23 | 29.76 | 43.83 | 72.83 |
| kitchen-mixed | 17.27 | 22.64 | 29.46 | 43.41 | 68.61 |
| *Average* | 17.71 | 22.33 | 29.21 | 43.95 | 70.94 |

