# OpenReview forum: "Moment Matching Q-Learning"
_ICML.cc/2026/Conference — ICML 2026 regular_

### Official Review · Reviewer_kLUf · 2026-02-28

**Soundness:** 3
**Presentation:** 4
**Significance:** 3
**Originality:** 3
**Overall Recommendation:** 4
**Confidence:** 3

**Summary:**

This paper proposes Moment Matching Q-Learning (MoMa QL), an offline(-to-online) reinforcement learning algorithm that integrates generative modeling with actor-critic training. The key idea is to learn an implicit action sampler based on stochastic interpolants and moment matching via Maximum Mean Discrepancy (MMD), enabling accelerated multi-step-to-single-step action inference along a probability flow trajectory. Experiments on D4RL benchmarks demonstrate convincing performance of MoMaQL.

**Compliance With Llm Reviewing Policy:**

Affirmed.

**Final Justification:**

After reading the rebuttal, I have decided to maintain my initial recommendation of weak accept.

**Key Questions For Authors:**

* If possible, can authors shou how much improvement in online RL performance is due to sampling speed vs improved policy quality?

**Limitations:**

The methodological and theoretical contributions appear relatively incremental, and the current theoretical analysis does not provide substantially new guarantees beyond existing frameworks.

**Strengths And Weaknesses:**

**Strengths**:
* This paper is well motivated: diffusion- and flow-based policies often suffer from slow sampling, which can be prohibitive in online or computationally intensive settings. The proposed “jump” mechanism across diffusion time steps is a reasonable attempt to reduce inference cost while retaining the expressivity of generative policies.
* The empirical experimental analysis is strong and broad.
* The presentation is clear and easy to follow.

**Weaknesses**:
* While the “moment matching jump” idea is interesting, the method appears to be a combination of lots of existing components.
* The empirical study is limited to D4RL benchmarks. Additional validation on larger-scale or real-world settings would strengthen the evidence for practical impact and generality.

---

> ### Author Rebuttal · Authors · 2026-03-31
>
> We appreciate the reviewer for their encouraging review and would like to address the questions from reviewers as below:
>
> **Q1.** While the “moment matching jump” idea is interesting, can you justify the technical novelty and challenges in MoMa QL?
>
> **A1.** Thank you for recognizing the uniqueness and significance of MoMa QL. We would like to clarify that MoMa QL is not a trivial combination of "moment matching jump" and regular Q-learning framework, but rather a synergistic integration that addresses fundamental bottlenecks in generative policy learning. Originally, moment matching jump was designed solely to accelerate the generation process. In our framework, we incorporate the Q-function directly into the generative guidance, dynamically shifting the target distribution to maximize expected returns. This integration of Q-learning into the moment-matching objective is, to the best of our knowledge, novel in the literature.
>
> Also, MoMa QL provides a more theoretically grounded and expressive policy class compared to other one-step or few-step generative policy. By introducing Maximum Mean Discrepancy (MMD), our method guarantees a deterministic minimizer under DDIM interpolant and a distribution-level convergence by matching all orders of statistics between the original and target distributions. In our work we also showed that the MMD loss could also be viewed as a natural generalization of the consistency training loss, which sorely matches the first-order and second-order moments and are susceptible to the influence of mode collapse as the data dimension increases.
>
> ---
> **Q2.** Can you add more experiments on larger-scale or real-world settings?
>
> **A2.** We sincerely thank the reviewer for the constructive suggestion. We fully agree that evaluating on larger-scale settings is crucial for demonstrating practical impact. To address this, we have extended MoMaQL's evaluation to OGBench, a recently proposed large-scale benchmark for offline RL that features high-dimensional observations and complex manipulation tasks (following the same evaluation protocol as FQL).
>
> While the full suite of experiments is still ongoing due to the computational scale of OGBench, we provide the results for several representative tasks below, ranging from basic manipulation to complex combinatorial puzzles:
>
> |Task|BC|CAC|IFQL|FQL|MomaQL|
> |--|--|--|--|--|--|
> |cube-single-play-singletask-task1-v0|10±5|77±28|79±4|97±2|**100±0**|
> |puzzle-3x3-play-singletask-task1-v0|5±2|97±2|94±3|90±4|**98±1**|
> |puzzle-4x4-play-singletask-task1-v0|1±1|44±10|49±9|34±8|**72±3**|
> |scene-play-singletask-task1-v0|19±6|100±1|98±3|100±0|**100±0**|
>
> ---
> **Q3.** How much improvement in online RL performance is due to sampling speed vs improved policy quality?
>
> **A3.** We thank the reviewer for pointing out this interesting perspective. To clarify, the improvements demonstrated in our online RL performance are fundamentally driven by the improved policy quality, while the accelerated sampling speed serves as the critical enabler that makes this online fine-tuning practically feasible. We would like to disentangle their contributions as follows:
>
> In standard RL evaluations, performance is measured against environment interaction steps rather than wall-clock time. Therefore, the significant performance gains originates from the robust and expressive policy representation enabled by MoMa QL. Also, the improvement in our policy also yields a robust and reliable Q function which could mitigate the out-of-distribution (OOD) estimation in the offline-to-online adaptation process, and thus provide a relatively stable offline-to-online learning curve.
>
> We also would like to highlight that this accelerated sampling speed, though not directly contributing to the performance gain, makes it feasible to transfer an offline diffusion policy to online within a reasonable training time. By accelerating action sampling, MoMa QL allows for simultaneous, high-frequency critic updates during policy training, bridging the gap between high-capacity generative models and real-time online reinforcement learning.

---

> > ### Author Rebuttal · Reviewer_kLUf · 2026-04-01
> >
> > Thank you for the helpful response. I have no further questions.

---

> > > ### Author Response · Authors · 2026-04-01
> > >
> > > Dear Reviewer kLUF,
> > >
> > > Thank you for your feedback and continuously support to our paper. We are glad to see our response have fully addressed your concern.
> > >
> > > Best regards,
> > > Authors.

---

### Official Review · Reviewer_cQFw · 2026-03-11

**Soundness:** 3
**Presentation:** 2
**Significance:** 3
**Originality:** 3
**Overall Recommendation:** 4
**Confidence:** 4

**Summary:**

This paper proposes a novel offline & offline-to-online reinforcement learning framework called Moment Matching Q-Learning (MoMa QL), which combines Maximum Mean Discrepancy (MMD) regularization with stochastic interpolants and a consistency-style training paradigm to accelerate sampling from score-based or flow-based generative policies.

* Problem: Although recent generative policies using score-based models and flow-based models demonstrate excellent expressiveness, their iterative sampling latency is computational bottleneck in reinforcement learning (RL), making them impractical.

* Solution: MoMa QL minimizes an MMD-based actor loss with a consistency-style self-consistency constraints (Eq. (10)) that matches marginal interpolant distributions, and enables fast sampling via stochastic interpolants with the marginal-preserving property.

* Theoretical contribution: The paper introduces a theoretically grounded actor loss that aligns denoising distributions through self-consistency and MMD matching at marginal interpolants to achieve higher-order moment alignment. Partial convergence guarantees for this loss are provided.

* Empirical results: Experiments on D4RL Gym, Adroit, and Kitchen benchmarks demonstrate that MoMa QL is competitive with or outperforms SOTA baselines across all three paradigms — Behavior Cloning (BC), Offline RL, and Offline-to-Online RL. In particular, MoMa QL achieves gains in training and inference speed through few-step sampling.

**Compliance With Llm Reviewing Policy:**

Affirmed.

**Final Justification:**

The rebuttal adequately addressed my primary concerns. I maintain my score and recommend acceptance.

**Key Questions For Authors:**

1. Performance in the Offline-to-Online Setting
> What is the authors' analysis of why the performance degradation was relatively small in the Offline-to-Online evaluation compared to other baselines?

2. Offline-to-Online Evaluation Protocol
> Is the Offline-to-Online evaluation protocol (100K pretraining followed by 400K fine-tuning in Line 305) a common choice, or was it selected based on a specific criterion?

**Limitations:**

Yes.

**Strengths And Weaknesses:**

1. Strength
* Soundness
> * The idea combining MMD, stochastic interpolants, and consistency is fresh and novel, and the paper provides theoretical analysis to support it rigorously.
> * The paper presents comprehensive experiments across major benchmarks, providing strong empirical support for the effectiveness of the proposed method.

* Presentation
> * Equations, graphs, and tables are effectively employed to support the reader's understanding.

* Significance
> * The paper addresses the sampling bottleneck in Diffusion/Flow-based RL policies, a timely and important problem in the field.

* Originality
> * Leveraging MMD at the conditional marginal level for learning generative policies in RL — especially when combined with marginal-preserving interpolants — represents a novel approach.

2. Weakness
* Soundness
> * The combination of double Q-learning and BC regularization in the value loss (Eq. (11)) is a well-established approach, and the novelty of the paper lies primarily in the actor loss term and the sampling acceleration component.

* Presentation
> * The abstract lack conciseness and contains several instances of awkward grammar and phrasing. For example, the description of the MMD as being used for statistical hypothesis testing appears unnecessary in this context. Additionally, "that intent" in Line 23 should be corrected to "that intends".
> * The text in the experimental results figures (Fig. 1-14) is too tiny and hard to read.

* Significance
> * It is encouraging that the MoMa QL exhibits minimal performance degradation due to catastrophic forgetting in the offline-to-online setting; however, the paper lacks a deeper analysis of why this is the case.

---

> ### Author Rebuttal · Authors · 2026-03-31
>
> We sincerely thank you for your positive feedback and detailed comments. We will incorporate your suggestions regarding the detailed presentation in our camera ready revision. We address the reviewer's questions below:
>
> **Q1.** How is the technical novelty of this paper?
>
> **A1.** We would like to highlight that the fundamental bottleneck in such policy regularization tasks inherently lies in the actor’s expressiveness, efficiency and robustness. MoMa QL has demonstrated superior empirical performance, fast inference speed and theoretically rigorous guarantees, and thus represents a great advancement to traditional generative policies.
>
> Following the same claim, MoMa QL also has the potential to be applied to control the generation process in other downstream tasks, including language models and computer vision. The generative policies in RL usually learn and sample from a conditional distribution, and its mathematical formulation is intrinsically identical to other conditional generation paradigms, including tasks such as conditional image generation and image inpainting. From this perspective, the core idea of MoMa QL could also be seamlessly extended to these tasks, offering an excellent solution to balance the performance and computational efficiency.
>
> ---
> **Q2.** Can you discuss more on the minimal performance degradation achieved by MoMa QL in offline-to-online setting?
>
> **A2.** We sincerely thank the reviewer for highlighting this positive aspect of MoMaQL and for encouraging a deeper analysis. The minimal performance degradation during the offline-to-online transition is indeed a core advantage of our method, which stems directly from how MoMaQL handles Q-value calibration. As discussed in Cal-QL, catastrophic forgetting typically occurs because traditional offline RL methods enforce strict conservatism, leading to severely underestimated Q-values. When transitioning online, the sudden exposure to ground-truth rewards causes these Q-values to shift drastically. Our method, MomaQL, ensures that the initial Q-values are closer to the true online scale.
>
> ---
> **Q3.** Is the Offline-to-Online evaluation protocol a common choice, or was it selected based on a specific criterion?
>
> **A3.** We confirm that use the same setting of epochs as mentioned in CAC [1], which is common in offline-to-online setting.
>
> [1] Ding, Zihan, and Chi Jin. "Consistency models as a rich and efficient policy class for reinforcement learning." arXiv preprint arXiv:2309.16984 (2023).

---

> > ### Author Rebuttal · Reviewer_cQFw · 2026-04-03
> >
> > We thank the authors for their detailed response and for addressing our questions.
> > We appreciate the authors’ clarification regarding the technical novelty of MoMa QL, particularly its potential applicability to other conditional generation tasks beyond reinforcement learning.
> >
> > We also thank the authors for providing a deeper analysis on the minimal performance degradation in the offline-to-online setting. The explanation concerning Q-value calibration and its relation to catastrophic forgetting is insightful and helps strengthen the paper.
> >
> > Additionally, we confirm that the Offline-to-Online evaluation protocol follows the setting used in CAC [1], which is acceptable. (2000 epochs, 20 tasks & 100k)
> >
> > Overall, the authors’ responses are satisfactory and have adequately addressed our main concerns. We believe the paper makes a solid contribution to the field.

---

> > > ### Author Response · Authors · 2026-04-04
> > >
> > > Dear Reviewer cQFW,
> > >
> > > Thank you for your feedback and continuous support. We are glad to see our response have fully addressed your concern.
> > >
> > > Best regards,
> > >
> > > The Authors Team

---

### Official Review · Reviewer_8Tcm · 2026-03-13

**Soundness:** 3
**Presentation:** 2
**Significance:** 3
**Originality:** 3
**Overall Recommendation:** 4
**Confidence:** 4

**Summary:**

The paper proposes Moment Matching Q-Learning, where a generative policy is learned in an Actor Critic framework.  The generative policy is implemented as DDIM interpolant , authors propose in this paper a behavior cloning (BC) loss that is tailored for this setup.  The behavior cloning loss uses a self consistency loss with the MMD aligning denoising distributions  at different times. A justification of this consistency is discussed in the appendix under marginal preserving interpolants. Authors show the efficiency of the method and superior performance superior performance in offline-to-online RL tasks.

**Compliance With Llm Reviewing Policy:**

Affirmed.

**Final Justification:**

I thank the authors for fixing the proof and providing that the assumption considered is realistic. I recommend accepting the paper.

**Key Questions For Authors:**

1- My main question how do you justify that the consistency loss indeed ensures behavior cloning? The proof under the assumption of marginal preserving interpolant seems not convincing.

2- did you monitor the MMD (p^theta_{s|t} (x|x_t) || q_{s})(x_s) )

3- line 137 and comment on moment for x high dimensional here x^j does not make sense .

4- since you are computing MMD in the kernel form  the cost will be quadratic in the number of points, have you thought of using random feature approximation, it will make MMD estimation linear : compute kernel mean embedding in feature space and l_2 in that feature space.

**Limitations:**

- The main limitations in the paper are the lack of clarity and the jumping notations
- Proofs in the appendix are not clear and hard to follow
- Lack of justification of the consistency loss for behavior cloning or experimental support for it

**Strengths And Weaknesses:**

* soundness: The paper is sound and presents a new behavior cloning loss for generative policies. The appendix justifying the loss in Section 4.3 is though hard to parse and the proof feels circular in the sense under the assumptions made , the result is obvious, but what i am missing is there anything in the construction or the learning that makes this assumption satisfied?

* presentation: The paper would benefit from a bit of refocus and rewrite to make it easier to follow.  For instance in Section 4 , the reader is not yet told that the policy will be stochastic interpolant , and the paper jumps between different notation $\pi_{\theta}$, $p_{\theta}$. Also the paper is not transparent about linking x, to action and state in the context of actor critic, and what is q(x_t) in this context, one can infer only things by looking at algorithm 3 in appendix. Also the choice of time s and state s is unfortunate , use **u** maybe.  The neural network G is mentioned in Section 4.3 without any further explanation. The paper would benefit from a diagram that would make the  paper digestable : show the sampling process in line 213-219  and your consistency loss.

* Significance: Authors obtain good performance and efficiency on benchmarks. The significance of the paper is overshadowed by the presentation that needs more work before the paper is ready for publication.

* Originality : Specializing consitency losses with MMD for generative policy is an interesting contribution.

---

> ### Author Rebuttal · Authors · 2026-03-31
>
> Thank you for your positive and careful review, we sincerely appreciate your comments in helping us polishing the presentation of the paper. Below we have addressed your questions and include a diagram per your suggestion.
>
> **Q1.** Can you clarify the justification of loss function defined in Section 4.3?
>
> **A1.** Thank you for pointing this out. We would like to elucidate it further: In Appendix A.2, we proved the existence of the deterministic minimizer and the convergence guarantee under the assumption of a marginal-preserving interpolant. Furthermore in Appendix A.3, we stated that this is satisfied if we utilize the DDIM interpolant in the reverse process. To clear up any concerns regarding circular reasoning, we provide a formal proof sketch verifying that the **DDIM interpolant intrinsically satisfies the marginal-preserving equality**
> > For a DDIM interpolant $x_t = I(x, \epsilon, t) = a_t x + b_t \epsilon$, the marginal distribution is $q_t(x_t) = \iint \delta(x_t - I(x, \epsilon, t)) q(x) p(\epsilon) dx d\epsilon$. In the DDIM framework, the trajectory is deterministic given $x$. Therefore, for any $s, t \in [0, 1]$, the state $x_s$ is uniquely determined by $x$ and $x_t$ and the conditional density is defined by $q_{s|t}(x_s | x, x_t) = \delta (x_s - [ a_s x + b_s (x_t - a_t x)/b_t )] )$.
>
> > We substitute the DDIM conditional $q_{s|t}$ and the joint distribution $q_t(x, x_t) = q_t(x | x_t) q_t(x_t)$ into the right-hand side (RHS) of Eq. (7): RHS$=\iint q_{s|t}(x_s | x, x_t) q_t(x, x_t) dx_t dx$, using the fact that $q_t(x, x_t) = \int \delta(x_t - I(x, \epsilon, t)) q(x) p(\epsilon) d\epsilon$, we expand the integral as RHS$= \iiint \delta ( x_s - [ a_s x + \frac{b_s}{b_t}(x_t - a_t x) ] ) \delta(x_t - (a_t x + b_t \epsilon)) q(x) p(\epsilon) d\epsilon dx_t dx$
>
> > Applying the fact that $a_s x + \frac{b_s}{b_t} ( (a_t x + b_t \epsilon) - a_t x ) = a_s x + b_s \epsilon = I(x, \epsilon, s)$, we have RHS$ = \iint \delta(x_s - I(x, \epsilon, s)) q(x) p(\epsilon) d\epsilon dx = q_s(x_s)$, which verifies that **DDIM interpolant intrinsically satisfies the marginal-preserving equality**
>
> This mathematically confirms that the marginal-preserving property is guaranteed by our algorithmic construction. We will explicitly include this proof in the revised appendix to ensure complete transparency.
>
> ---
> **Q2.** Did you monitor the MMD$(p^\theta_{s|t} (x|x_t) || q_{s}(x_s))$
>
> **A2.** Yes, we tracked the MMD loss in empirical form as mentioned in Appendix A.3. However, we observed that monitoring this sampling loss provides limited insight into the overall policy performance. The MMD loss converges extremely fast during the initial stages of training, after which its rate of decrease becomes marginal. This phenomenon also appears in the baseline algorithms such as Diffusion-QL and FQL. We put the results as a csv file in our anonymous link: https://anonymous.4open.science/r/ICML_rebuttal-abcdEFGH.
>
> ---
> **Q3.** What does $x^j$ mean in Line 137?
>
> **A3.** Here the superscript $j$ denotes the order of the moment rather than element-wise exponential symbol, and $x^j$ actually denotes a rank-$j$ tensor. For example, when $j = 1$, $E_\mu[x]$ is the first order moment; when $j = 1$, $E_\mu[x^2] = E_\mu[xx^T]$ is the second order moment; For $j \geq 3$, it naturally extends to higher-order moment tensors. In the revised manuscript, we will explicitly define this tensor notation to prevent any mathematical misunderstanding. Nevertheless, the definition of $x^j$ is just for the preliminary information for RKHS and does not explicitly used in our results.
>
> ---
> **Q4.** Is it possible to use random feature approximation to linearize the MMD kernel from $O(N^2)$ to $O(N)$.
>
> **A4.** We sincerely thank the reviewer for this highly insightful suggestion which could maximize the efficiency of computing the MMD loss. Although our work mostly consider utilizing the MMD in the context of RL, we will incorporate a paragraph discussing this direction in the future works.
>
> ---
> **Q5.** Can you incorporate a diagram to help digest the paper?
>
> **A5.** Thank you for the suggestion, we have incorporated a diagram for the main algorithm of MoMa QL in our anonymous link: https://anonymous.4open.science/r/ICML_rebuttal-abcdEFGH.

---

> > ### Author Rebuttal · Reviewer_8Tcm · 2026-04-01
> >
> > Thank you for the clarification. I did not check carefully the proof sketch you provided here, but if this is correct this would resolve the circular issue in the proof indeed.

---

> > > ### Author Response · Authors · 2026-04-01
> > >
> > > Dear Reviewer 8Tcm,
> > >
> > > We sincerely thank you for your feedback and keeping the positive feedback. We have double checked carefully that the proof is correct since it's following the standard derivation of the DDIM analysis. Therefore we believe the circular issue should be addressed. We will put this paragraph into the proof in our revision.

---

### Official Review · Reviewer_Q912 · 2026-03-14

**Soundness:** 4
**Presentation:** 2
**Significance:** 3
**Originality:** 3
**Overall Recommendation:** 4
**Confidence:** 3

**Summary:**

This paper presents Moment Matching Q-Learning (MoMa QL), a mean-field flow matching appraoch for modeling the trajectory distribution, solving the sample speed bottleneck for decision-making. By learning a function mapping from any marginal distribution at time $t$ to any marginal at time $s < t$, this framework facilitates a seamless transition across the probability flow trajectory. The extensive experiments on D4RL, gym, and offline-online finetuning show the effectiveness of this approach.

**Compliance With Llm Reviewing Policy:**

Affirmed.

**Final Justification:**

I thank authors' efforts in the rebuttal. First, they clarifed the key difference between them and one-step flow matching methods, which is critical in evaluating their novelty. Second, their additional experiments further demonstrated the soundness.

**Key Questions For Authors:**

See Weakness

**Limitations:**

Yes

**Strengths And Weaknesses:**

Strengths:

(1) The mean-field approach solves the sampling speed bottleneck, which is critical for online decision-making application.

(2) The experiments are comprehensive, including various benchmarks, as well as additional offline-online study.

(3) The paper is well written.

Weaknesses:

(1) What's the key difference between MoMa QL and [1] ? It seems like a simple application of [1] to decision-making.

(2) In Table 1, among 3 medium-expert （me）datasets, Moma QL falls behind other baselines in 2 of them. Can authors give more explanations on it?

(3) In Figure 1, same environment but different expert-level datasets, why for the right subfigure there is a drop-down in the middle while the left doesn't have it? More disscussion is encouraged.

Reference:

[1] Geng, Zhengyang, Mingyang Deng, Xingjian Bai, J. Zico Kolter, and Kaiming He. "Mean flows for one-step generative modeling." arXiv preprint arXiv:2505.13447 (2025).

---

> ### Author Rebuttal · Authors · 2026-03-31
>
> Thank you for your feedback, we address your questions as below:
>
> **Q1.** What's the key difference between MoMa QL and MeanFlow [1]
>
> **A1.** Thank you for pointing that related work. In particular, MeanFlow achieves one-step and few-steps sampling by utilizing the Jacobian Vector Production and stop-gradient trick to train the model. However, as author of DM1[1] explicitly notes, directly porting MeanFlow into RL will introduce representation collapse, which is also common for other accelerated generative policies such as CAC[2]. Instead, our method focuses more on learning a better policy distribution via matching all orders of the moments between the expert and learned policy via MMD loss and a deliberately chosen kernel. By mapping the distributions in a Reproducing Kernel Hilbert Space (RKHS), our algorithm has robust statistical properties and a more theoretically grounded representation, which could be beneficial to prevent mode collapse and mitigate the OOD overestimation. So, while both MoMa QL and MeanFlow aim to achieve efficient generative modeling.
>
> ---
> **Q2.** Why Moma QL falls behind other baselines in medium-expert datasets?
>
> **A2.** Thank you for this insightful observation. Our insight is that the medium-expert datasets contain a high proportion of optimal trajectories. In such scenarios, existing baselines can easily extract and exploit this high-quality *expert data* within the in-distribution dataset. In contrast, MoMaQL is fundamentally designed to tackle more challenging and diverse environments. Our method maintains highly competitive performance on these less challenging medium-expert tasks, while significantly outperforming existing baselines in more complex, sub-optimal settings. We believe this demonstrates the robust generalization capability of MoMaQL.
>
> ---
> **Q3.** Can you comment more on why the difference performance drop in offline-to-online setting in Figure 1?
>
> **A3.**  Thank you for this question, we find that the temporary drop in Figure 1(b) stems from the Q-value scale mismatch during the transition from conservative offline initialization to online interaction. In particular, such a performance drop becomes more significant in `medium` dataset comparing with the `medium-replay` dataset. We hypothesis that this is because the `medium` dataset lack essential support to the near-optimal policy since it is composed by collecting the medium-performance policies. In contrast, `medium-replay` dataset contains more diverse trajectories leveraging the replay buffer through online interaction.  Following the analysis in Cal-QL, conservative offline RL often underestimates Q-values to a point where they no longer provide a well-calibrated ranking for online exploration. To further demonstrate that this transient performance dip is a consistent and theoretically-grounded phenomenon during the O2O transition, we have compiled additional experimental results across multiple D4RL tasks:
>
> |Task|Offline|Online|
> |--------|:--------:|:-------:|
> |halfcheetah-medium-expert-v2|52.9|61.3|
> |halfcheetah-medium-replay-v2|54.8|62.2|
> |halfcheetah-medium-v2|69.9|52.2|
> |hopper-medium-expert-v2|20.5|11.3|
> |hopper-medium-replay-v2|98.9|76.8|
> |hopper-medium-v2|22.6|10.1|
> |walker2d-medium-expert-v2|4.2|16.8|
> |walker2d-medium-replay-v2|82.4|93.0|
> |walker2d-medium-v2|7.7|9.3|
>
> [1]. Zou, G., Wang, H., Wu, H., Qian, Y., Wang, Y., & Li, W. (2025). DM1: MeanFlow with Dispersive Regularization for 1-Step Robotic Manipulation. arXiv preprint arXiv:2510.07865
>
> [2]. Ding, Z., & Jin, C. (2023). Consistency models as a rich and efficient policy class for reinforcement learning. arXiv preprint arXiv:2309.16984.

---

> > ### Author Rebuttal · Reviewer_Q912 · 2026-04-03
> >
> > Dear Authors,
> >
> > Thank you for the efforts answering my questions, and they are all solved. I will raise my overall score.

---

> > > ### Author Response · Authors · 2026-04-04
> > >
> > > Dear Reviewer Q912,
> > >
> > > Thank you for your feedback and glad to see your questions have been resolved.  We also appreciate greatly that you raised the score. We remain at your disposal for any further clarifications you may require.
> > >
> > > Best regards,
> > > The Authors Team

---

### Decision · Program_Chairs · 2026-04-30

**Decision:**

Accept (regular)

**Comment:**

The committer recommends acceptance. The paper addresses a critical bottleneck in the deployment of generative policies for RL—inference speed—using a theoretically grounded moment-matching objective. The inclusion of additional large-scale results from OGBench and the clarification of the marginal-preserving property of the DDIM interpolant were particularly appreciated and helped move the consensus toward acceptance.

As you prepare your camera-ready version, please pay close attention to the presentation issues raised.A more explicit discussion in the main text regarding the "representation collapse" you observed when using simpler flow-matching techniques would also greatly benefit future researchers.